# MOTIF-DRIVEN CONTRASTIVE LEARNING OF GRAPH REPRESENTATIONS

## ABSTRACT

Graph motifs are significant subgraph patterns occurring frequently in graphs, and they play important roles in representing the whole graph characteristics. For example, in chemical domain, functional groups are motifs that can determine molecule properties. Mining and utilizing motifs, however, is a non-trivial task for large graph datasets. Traditional motif discovery approaches rely on exact counting or statistical estimation, which are hard to scale for large datasets with continuous and high-dimension features. In light of the significance and challenges of motif mining, we propose *MICRO-Graph*: a framework for MotIf-driven Contrastive leaRning Of Graph representations to: 1) pre-train Graph Neural Networks (GNNs) in a self-supervised manner to automatically extract motifs from large graph datasets; 2) leverage learned motifs to guide the contrastive learning of graph representations, which further benefit various downstream tasks. Specifically, given a graph dataset, a motif learner cluster similar and significant subgraphs into corresponding motif slots. Based on the learned motifs, a motif-guided subgraph segmenter is trained to generate more informative subgraphs, which are used to conduct graph-to-subgraph contrastive learning of GNNs. By pre-training on ogbg-molhiv molecule dataset with our proposed *MICRO-Graph*, the pre-trained GNN model can enhance various chemical property prediction downstream tasks with scarce label by 2.0%, which is significantly higher than other state-of-the-art self-supervised learning baselines.

## 1 INTRODUCTION

Graph-structured data, such as molecules and social networks, is ubiquitous in many scientific research areas and real-world applications. To represent graph characteristics, graph motifs were proposed in Milo et al. (2002) as significant subgraph patterns occurring frequently in graphs and uncovering graph structural principles. For example, functional groups are important motifs that can determine molecule properties. Like the hydroxide ($-OH$) usually implies higher water solubility, and for proteins, Zif268 can mediate protein-protein interactions in sequence-specific DNA-binding proteins. (Pabo et al., 2001).

Graph motifs has been studied for years. Meaningful motifs can benefit many important applications like quantum chemistry and drug discovery (Ramsundar et al., 2019). However, extracting motifs from large graph datasets remains a challenging question. Traditional motif discovery approaches (Milo et al., 2002; Kashtan et al., 2004; Chen et al., 2006; Wernicke, 2006) rely on discrete counting or statistical estimation, which are hard to generalize to large-scale graph datasets with continuous and high-dimension features, as often the case in real-world applications.

Recently, Graph Neural Networks (GNNs) have shown great expressive power for learning graph representations without explicit feature engineering (Kipf & Welling, 2016; Hamilton et al., 2017; Veličković et al., 2017; Xu et al., 2018). In addition, GNNs can be trained in a self-supervised manner without human annotations to capture important graph structural and semantic properties (Veličković et al., 2018; Hu et al., 2020c; Qiu et al., 2020; Bai et al., 2019; Navarin et al., 2018; Wang et al., 2020; Sun et al., 2019; Hu et al., 2020b). This motivates us to rethink about motifs as more general representations than exact structure matches and ask the following research questions:

- Can we use GNNs to automatically extract graph motifs from large graph datasets?
- Can we leverage the learned graph motif to benefit self-supervised GNN learning?

In this paper, we propose *MICRO-Graph*: a framework for MotIf-driven Contrastive leaRning Of Graph representations. The key idea of this framework is to learn graph motifs as prototypical cluster centers of subgraph embeddings encoded by GNNs. In this way, the discrete counting problem is transfered to a fully-differentiable framework that can generalize to large-scale graph datasets with continuous and high-dimensional features. In addition, the learned motifs can help generate more informative subgraphs for graph-to-subgraph contrastive learning. The motif learning and contrastive learning are mutually reinforced to pre-train a more generalizable GNN encoder.

For motif learning, given a graph dataset, a motif-guided subgraph segmenter generates subgraphs from each graph, and a GNN encoder turns these subgraphs into vector representations. We then learn graph motifs through clustering, where we keep the K prototypical cluster centers as representations of motifs. Similar and significant subgraphs are assigned to the same motif and become closer to their corresponding motif representation. We train our model in an Expectation-Maximization (EM) fashion to update both the motif assignment of each subgraph and the motif representations.

For leveraging learned motifs, we propose a graph-to-subgraph contrastive learning framework for GNN pre-training. One of the key components for contrastive learning is to generate semantically meaningful views of each instance. For example, a continuous span within a sentence (Joshi et al., 2020) or a random crop of an image (Chen et al., 2020). For graph data, previous approaches leverage node-level views, which is not sufficient to capture high-level graph structural information Sun et al. (2019). As motifs can represent the key graph properties by its nature, we propose to leverage the learned motifs to generate more informative subgraph views. For example, alpha helix and beta sheet can come together as a simple $\beta\beta\alpha$ fold to form a zinc finger protein with unique properties. By learning such subgraph co-occurrence via contrastive learning, the pre-trained GNN can capture higher-level information of the graph that node-level contrastive can't capture.

The pre-trained GNN using *MICRO-Graph* on the ogbg-molhiv molecule dataset can successfully learn meaningful motifs, including Benzene rings, nitro, acetate, and etc. Meanwhile, fine-tune this GNN on seven chemical property prediction benchmarks yielding 2.0% average improvement over non-pretrained GNNs and outperforming other self-supervised pre-training baselines. Also, extensive ablation studies show the significance of the learned motifs for the contrastive learning.

## 2 RELATED WORK

The goal of self-supervised learning is to train a model to capture significant characteristics of data without human annotations. This paper studies whether we can use such approach to automatically extract graph motifs, i.e. the significant subgraph patterns, and leverage the learned motifs to benefit self-supervised learning. In the following, we first review graph motifs especially challenges for motif mining, and then discuss approaches for pre-training GNNs in a self-supervised manner.

**Graph motifs** are building blocks of complex graphs. They reveal the interconnections of graphs and represent graph characteristics. Mining motifs can benefit many tasks from exploratory analysis to transfer learning (Henderson et al., 2012). For many years, various motif mining algorithms have been proposed. There are generally two categories, either exact counting as in Milo et al. (2002); Kashtan et al. (2004); Schreiber & Schwöbbermeyer (2005); Chen et al. (2006), or sampling and statistical estimation as in Wernicke (2006). However, both approaches cannot scale to large graph datasets with high-dimension and continuous features, which is common in real-world applications. In this paper, we proposes to turn the discrete motif mining problem into a GNN-based differentiable cluster learning problem that can generalize to large-scale datasets. Another GNN-based work related to graph motifs is the GNNExplainer, which focuses on post-process model interpretation(Ying et al., 2019). It can identify substructures that are important for graph property prediciton, e.g. motifs. The difference between GNNExplainer and MICRO-Graph is that the former identify motifs at a single graph level, and the later learns motifs across the whole dataset.

**Contrastive learning** is one of the state-of-the-art self-supervised representation learning algorithms. It achieves great results for visual representation learning (Chen et al., 2020; He et al., 2019). Contrastive learning forces views generated from the same instance (e.g. different crops of the same image) to become closer, while views from different instances apart. One key component in contrastive learning is to generate informative and diverse views from each data instance. In computer vision, researchers use various techniques, including cropping, color distortion, and Gaussian

blurs to generate views. However, when it comes to graphs, constructing informative view of graph is a challenging task. In our framework, we utilize the learned motifs, which are significant subgraph patterns, to guide view (subgraph) generation, and conduct graph-to-subgraph contrastive learning.

**Self-supervised learning for GNNs** also draws many attention recently. For graphs, representations can be at different levels, e.g. node level and (sub)graph level. Veličković et al. (2018); Hu et al. (2020c); Qiu et al. (2020) mainly focus on node-level representation learning in a single large graph, as opposed to the focus of this paper, which is representation learning of whole graphs. Hu et al. (2020b) provides a systematic analysis of pre-training strategies on graphs for both node-level and graph-level. However, only the node-level learning is self-supervised, and annotated labels are utilized for supervised learning at the graph level. For graph level self-supervised representation learning, Sun et al. (2019) proposed a contrastive framework, InfoGraph, to maximize the mutual information between graph representations and node representations. In Rong et al. (2020), the GROVER model and a motif based self-supervised learning task was proposed, where the discrete motifs are first extracted using a professional software, and then these motifs are used as prediction labels for pre-training the model. The difference between motifs in GROVER and in MICRO-Graph is that GROVER uses discrete structures, but MICRO-Graph uses continuous vector embeddings To alleviate these issues, we propose graph-to-subgraph view self-supervised contrastive learning, and the subgraph generation is guided by the learned motifs.

## 3 METHODOLOGY

The goal of this paper is to train a GNN encoder that can automatically extract graph motifs, i.e. significant subgraph patterns. Motif discovery on discrete graph structures is a combinatorial problem, and it is hard to generalize to large datasets with continuous features. We thus propose to formalize this problem as a differentiable clustering learning problem and solve it via self-supervised GNN learning. In this section, we formalize the problem and introduce the overall framework of *MICRO-Graph* in Section 3.1, and then describe each module in details in the following sections.

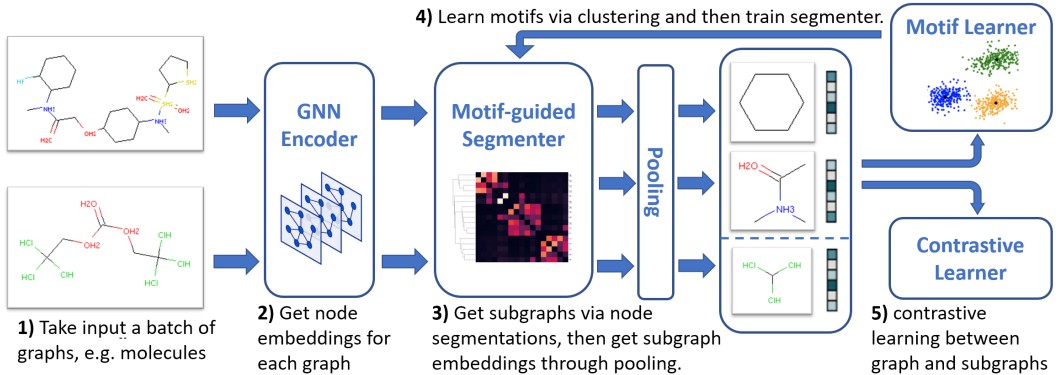

Figure 1: Overall framework of *MICRO-Graph*. A GNN trained in a self-supervised manner to automatically extract motifs. The learned motifs are leveraged to generate informative subgraphs for graph-to-subgraph contrastive learning.

### 3.1 THE OVERALL FRAMEWORK OF *MICRO-Graph*

Given a dataset with M graphs $\mathbb{G} = \{\mathcal{G}_1, ..., \mathcal{G}_M\}$, the differentiable clustering learning problem is meant to learn two things. One is a GNN-based graph encoder $\boldsymbol{E}(\cdot)$ that maps input (sub)graphs an embedding vector. The other is a K-slot embedding table $\{\boldsymbol{m}_1, ..., \boldsymbol{m}_K\}$, where each slot is a motif vector $\boldsymbol{m}$ corresponding to a cluster center of embeddings of frequently occurred subgraphs. To tackle this problem, we introduce the *MICRO-Graph* framework which consists three modules: 1) a *Motif-guided Segmenter* to extract important subgraphs; 2) a *Motif-Learner* to cluster sampled subgraphs and identify motifs; 3) a constrastive learning module for graph-to-subgraph contrastive learning. The overall framework is shown in Figure1. We describe details of each module in the

following sections. The *Motif Learner* is introduced first in Section 3.2, and then the *Motif-guided Segmenter* and the constrastive learning module in Section 3.3 and 3.4 respectively.

## 3.2 MOTIF LEARNER VIA EM CLUSTERING

The *Motif Learner* learns motifs by applying an Expectation-Maximization (EM) style clustering algorithm on sampled subgraphs. To start clustering, the *Motif-guided Segmenter* first extracts $N$ subgraphs $\{g_j\}_{j=1}^N$ from the input whole graphs. For each subgraph $g_j$, we generate its embedding $e_j = E(g_j)$ and calculate the cosine similarity between $e_j$ and each of the K motif vectors $\{m_1, ..., m_K\}$. We denote the similarity between $e_j$ and the kth motif vector as $S_{k,j} = \phi(m_k)^T \phi(e_j)^1$. In vector notation, the K-dimensional vector of similarities between $g_j$ and all K motif vectors is denoted as $s_j$, and the K-by-N-dimensional motif-to-subgraph similarity matrix is denoted as $S$, where the $j$-th column of $S$ is $s_j$ and the entry (k, j) of $S$ is $S_{k,j}$.

**E-Step.** The goal of the E-step is to come up with motif-based cluster assignments for subgraph embeddings $\{e_j\}_{j=1}^N$. The assignments can be represented by a K-by-N-dimensional matrix $Q = [q_1, ..., q_N]$, where the $j$-th column $q_j$ contains the probabilities of assigning the $j$-th subgraph to each of the K motifs. Each $q_j$ can be a one-hot vector for hard clustering or a probability vector with all the entries sum up to one for soft-clustering. This vanilla version clustering problem boils down to maximizing the objective $Tr(Q^T S)$, which corresponds to an assignment $Q$ that maximizes similarities between embeddings and its assigned motif. This objective works fine for a traditional EM clustering algorithm when embeddings are fixed. However, since representations will change when doing representation learning, this vanilla objective in the E-step can lead to a degenerate solution, i.e. all representations collapse to a single cluster center. To avoid this issue, we follow YM. et al. (2020) to introduce an entropy term and an equal-size constraint on $Q$ for clusters to have similar sizes. Our final objective is:

$$\max_{Q \in \mathcal{Q}} Tr(Q^T S) + \frac{1}{\lambda} H(Q) \tag{1}$$

where $H(Q) = -\sum_{i,j} Q_{i,j} \log Q_{i,j}$ is the entropy function, and the constraint set $\mathcal{Q}$ requires the marginal projection of $Q$ onto its columns and rows to be uniform.

$$\mathcal{Q} = \{Q \in \mathbb{R}_+^{K,N} | Q \mathbf{1}_N = \frac{\mathbf{1}_K}{K}, Q \mathbf{1}_K = \frac{\mathbf{1}_N}{N}\} \tag{2}$$

where $\mathbf{1}_N$ and $\mathbf{1}_K$ are all one vectors. This constraint optimization problem turns out to be an optimal transportation problem with a closed-form solution as (3) and can be solved efficiently using a fast Sinkhorn-Knopp algorithm.

$$Q^* = diag(u) \cdot \exp(\lambda S) \cdot diag(v) \tag{3}$$

Here $u$ and $v$ are normalization vectors. The derivations can be found in Cuturi (2013).

**M-Step.** The goal of the M-step is to maximize the log-likelihood of our data given the cluster assignment matrix $Q$ estimated in the E-step. We update parameters in the GNN encoder and the motif embedding table through the M-step. This step is equivalent to a supervised K-class classification problem with labels $Q$ and prediction scores $S$. Thus, we first apply a columnwise softmax normalization with temperature $\tau_g$ to $S$ to convert all entries of $S$ to probabilities, i.e. $\tilde{S}_{k,j} = \text{softmax}_k(S_{k,j}/\tau_g)$. Then we use the negative likelihood as the loss function.

$$\mathcal{L}_m = -\frac{1}{N} \sum_{j=1}^N \sum_{k=1}^K Q_{k,j} \log \tilde{S}_{k,j} \tag{4}$$

## 3.3 MOTIF-GUIDED SUBGRAPH SEGMENTER

Sampling informative subgraphs is crucial for both the *Motif-Learner* and the contrastive learning module . Traditional heuristic approaches such as random walk and k-hop neighbour sampling cannot guarantee to generate semantically reasonable and informative subgraphs. For example, heuristically sampled molecule subgraphs are likely to be a chain of carbons, which doesn't contain much

---

[1] For notation simplicity, we denote a L-2 normalization operator $\phi(\cdot)$ such that $\phi(x) = x / \|x\|_2$.

information about the original molecule, or it can be a fragment of a meaningful chemical structure, which loses the original chemical property. Since motifs are by nature informative subgraph patterns, we propose to leverage the learned graph motifs to design the *Motif-guided segmenter*, which will learn to segment a given graph into several subgraphs that are close to some discovered motifs.

**Subgraph Sampling via Segmentation.** To generate subgraphs from a graph $\mathcal{G}_i$ via segmentation, we first generate a node affinity matrix $\boldsymbol{A}^{(i)}$, and then do segmentation based on $\boldsymbol{A}^{(i)}$. Specifically, given the graph $\mathcal{G}_i$ with $n$ nodes, we use the GNN encoder $\boldsymbol{E}(\cdot)$ to generate the D-dimensional node embeddings $\{\boldsymbol{n}_1, ..., \boldsymbol{n}_n\}$ of all n nodes in $\mathcal{G}_i$. After that, we compute the n-by-n-dimensional node affinity matrix $\boldsymbol{A}^{(i)}$ by first computing pairwise cosine similarities between node embeddings, and then applying row-wise softmax normalization with temperature $\tau_n$ to the cosine similarity matrix. This normalization step is important because it transforms all the affinity scores to be in the range (0,1) and make the affinity scores of node pairs with high cosine similarities further stand out. The formula for computing the affinity score between node $s$ and node $t$ in graph $\mathcal{G}_i$, i.e. the entry $(s,t)$ of $\boldsymbol{A}^{(i)}$ is the following.

$$A_{s,t}^{(i)} = \mathrm{softmax}_s\big(\phi(\boldsymbol{n}_s)^T \phi(\boldsymbol{n}_t)/\tau_n\big) \tag{5}$$

Afterwards, we treat this affinity matrix $\boldsymbol{A}^{(i)}$ as a complete graph with n nodes and affinity scores as edge weights. Applying spectral clustering on this complete graph segments nodes into different groups. Within these groups, the connected components that have more than three nodes are collected as our sampled subgraphs. Multiple subgraphs may be sampled from the whole graph $\mathcal{G}_i$. For a particular subgraph $g_j$, its embedding $\boldsymbol{e}_j$ will be generated by indexing and aggregating the node embeddings generated when computing the affinity matrix. For example, if $\mathbb{I}$ is the indices of the nodes forming the subgraph $g_j$ selected by the *Motif-guided segmenter*, then $\boldsymbol{e}_j = Aggregate(\{\boldsymbol{n}_1, ..., \boldsymbol{n}_n\}[\mathbb{I}])$. The aggregate operation can be any order-invariant operation over a set of vectors, e.g. mean, sum, or elementwise max. In our experiment, we follow the start-of-the-art result from previous works and use the mean. Collecting subgraphs from all $M$ whole graphs result in the total set of subgraphs $\{g_j\}_{j=1}^N$ mentioned above.

**Motif-guided Training.** To train the segmenter to produce subgraphs close to motifs, the motif-to-subgraph similarity matrix $\boldsymbol{S}$ is used as the guidance. For a subgraph $g_j$ sampled from a whole graph $\mathcal{G}_i$ whose node affinity matrix is $\boldsymbol{A}^{(i)}$. If the similarity between $g_j$ and any motif is higher than a threshold, we make the affinity values among all the nodes within $g_j$ to increase, and the affinity value between these nodes and other nodes not in $g_j$ to decrease. The loss function is as below.

$$\mathcal{L}_s = -\frac{1}{N}\sum_{i=1}^N \sum_{(s,t)\in g_j} A_{s,t}^{(i)} \cdot \mathbf{1}\{\exists k \mid S_{k,j} > \eta_k, \forall 1 \le k \le K\} \tag{6}$$

Here $\eta_k$ is the threshold used to decide whether a subgraph is similar enough to the learned mofit k, and it is dynamically computed. In each iteration, we set $\eta_k$ to select the top 10% most similar subgraphs to motif k. The intuition is that if the subgraph $g_j$ is considered similar to a motif, then we update the embeddings of its nodes to become similar. By optimizing this loss, during the next sampling round, nodes produced motif-like subgraphs are more likely to be segmented together, which leads to more subgraph samples align with the motifs.

### 3.4 CONTRASTIVE LEARNING BETWEEN GRAPHS AND SUBGRAPHS

An expressive GNN encoder $\boldsymbol{E}(\cdot)$ is essential for capturing graph properties and accurately identifying motifs. We thus introduce a constrastive learning module to help the GNN learning. This module and the *Motif Learner* will mutually enhance each other to train a better GNN.

Contrastive learning is one of the state-of-the-art self-supervised learning methods. One key component in contrastive learning is to generate informative and diverse views of data instances. Previous contrastive methods on graphs utilized either nodes or whole graphs as views, which do not very well capture the micro-structure of graphs. To alleviate this issue in our constrastive learning module, we use the subgraph generated by our *Motif-guided segmenter* as one view of the graph, and the whole graph as another view.

Similarly to how we generate the subgraph embeddings, the whole graph embedding $\boldsymbol{h}_i$ of a graph $\mathcal{G}_i$ is generated by aggregating node embeddings output by the GNN encoder $\boldsymbol{E}(\cdot)$, $\boldsymbol{h}_i = Aggregate(\{\boldsymbol{n}_1, ..., \boldsymbol{n}_n\})$. Then we construct the M-by-N dimensional graph-to-subgraph similarity matrix $\boldsymbol{W}$, which is cosine similarity between each graph-subgraph pair followed by a row-wise softmax normalization with temperature $\tau_g$.

$$W_{i,j} = \text{softmax}_i\big(\phi(\boldsymbol{h}_i)^T \phi(\boldsymbol{e}_j)/\tau_g\big) \tag{7}$$

For the whole graph $\mathcal{G}_i$, subgraphs sampled from it are considered as positive pairs to it, and subgraphs sampled from other graphs are considered as negative pairs to it. Then the contrastive objective function is the following:

$$\mathcal{L}_c = -\frac{1}{M} \sum_{i=1}^{M} \sum_{j=1}^{N} W_{i,j} \cdot \mathbf{1}\{g_j \in \mathcal{G}_i\} \tag{8}$$

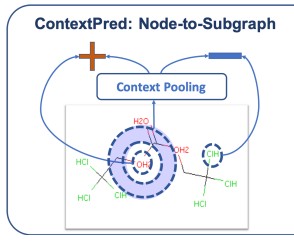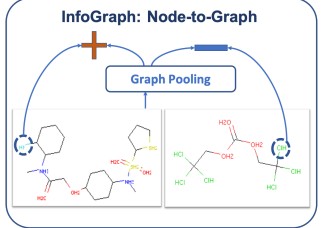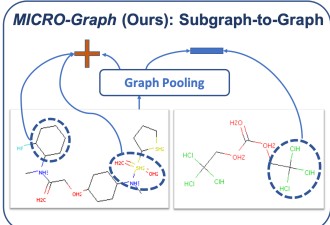

Figure 2: (Figure updated) Different ways to generate views for graph data. Context Prediction uses one node and one context graph as views. InfoGraph uses one node and the whole graph as views. Our method uses a higher-level subgraph as one view and the whole graph as another; thus the GNN can capture more global contextual information.

We show the view-generation difference between our framework and other existing methods in Figure 2. Existing contrastive learning methods rely on node-node views or node-graph views. Researchers in computer vision domain have found that conducting contrastive learning on more flexible views, e.g. arbitrary-sized image crops, leads to better representations, especially much better than lower pixel-level views (Chen et al., 2020). In our framework, the contrastive learning on graph-subgraph views is similar to this intuition from computer vision. It can capture higher-level information the node views cannot capture, and thus produce more meaningful representations.

Note that the subgraphs we utilize is generated by our *Motif-guided Segmenter*. In section 4, we systematically study the influence of subgraph sampling methods. Experiment results show that simple heuristic sampling methods lead to bad generalization performance. This shows the significance of leveraging the learned graph motifs to generate more informative views of graph-structured data.

### 3.5 JOINT TRAINING

The overall objective of *MICRO-Graph* is a weighted sum of all three loss terms described above.

$$\mathcal{L} = \lambda_m \mathcal{L}_m + \lambda_s \mathcal{L}_s + \lambda_c \mathcal{L}_c \tag{9}$$

With the proper constraints introduced to the E-step of the *Motif-Learner* (equation (1) and (2)), the whole framework can be trained end-to-end without worrying about degenerate solutions.

*MICRO-Graph* can simultaneously train an expressive GNN encoder and learn motifs of the given graph dataset. Moreover, the motif learning and contrastive learning are mutually reinforced: a better GNN produces more accurate subgraph embeddings and thus help motif mining, while better motifs can help generate more informative graph-to-subgraph views and benefit contrastive learning.

We show the the pseudocode of our approach in Algorithm 1. First, initialize the motif vectors and GNN encoder (line 2 - 3). For each batch of graphs $\mathbb{G}$, our segmenter will calculate the node-node affinity matrices $\boldsymbol{A}$ and extract subgraphs based on the affinity scores (line 5 - 6). After that, we apply GNN message passing on whole graphs to get the node embeddings and aggregate them for

both graph and subgraph embeddings (line 7 - 9). The next step is to compute the motif-to-subgraph similarity matrix $S$. Using $S$ we compute both the threshold $\eta$ and cluster assignment matrix $Q$ (line 11 - 13). With all these values, the three loss terms and the final joint loss introduced above can be computed (line 15 - 20).

**Algorithm 1** Pseudocode of MICRO-Graph in PyTorch Style, full version in Appendix A

```
1   # temperature parameters: tau_g, tau_n, weight parameters: lamb_m, lamb_c, lamb_s
2   model = Motif(args) # model contains all motif vectors, model.motifs
3   encoder = GNN(args) # GNN encoder for pre-training
4   for G in loader:
5       #I: node index of subgraph,  A: affinity matrices, num_subs: # of subgraphs per graph
6       I, num_subs, A = segmenter(G, encoder)
7       n = encoder(G)
8       h = aggregate(n, G) # M x D
9       e = aggregate(n, G, I) # N x D
10      S = pairwise_cosine_sim(model.motifs, e) # K x N
11      with torch.no_grad():
12          eta = topk_threshold(S) # take the topk similarity scores
13          Q = sinkhorn(S) # use the Sinkhorn-Knopp to solve for Q
14
15      loss_s = sampler_loss(A, eta)
16      S_tilde = torch.softmax(S / tau_g, dim=0) # K x N
17      loss_m = motif_loss(Q, S_tilde)
18      W = torch.softmax(pairwise_cosine_sim(h, e)/tau_g, dim=1)
19      loss_c = contrastive_loss(W, num_subs) # need num_subs from segmenter
20      loss = lamb_m*loss_m + lamb_c*loss_c + lamb_s*loss_s
```

# 4 EXPERIMENTS

We evaluate the effectiveness of *MICRO-Graph* from two perspective: 1) Whether the self-supervised framework can learn better GNNs that generalize well on graph classification tasks; 2) whether the learned motifs are reasonable and can truly benefit contrastive learning.

We mainly focus on chemical property prediction tasks. Specifically, we pre-train GNNs using *MICRO-Graph* on the ogbg-molhiv dataset from Open Graph Benchmark (OGB) (Hu et al., 2020a). This dataset contains 40K molecules. We test our pre-trained model on smaller molecule graph classification datasets. For more details of the datasets, please see Appendix E.

## 4.1 EVALUATION PROTOCOLS

We evaluate the effectiveness of pre-training using the following two evaluation protocols.

**Transfer Learning Setting**: we fine-tune the pre-trained GNN model with a small portion of labels on downstream tasks. We adopt the same train-test and model selection procedure as in Yanardag & Vishwanathan (2015); Zhang et al. (2018); Xu et al. (2018), where we perform 10-fold cross-validation and select the epoch with the best cross-validation performance averaged over the 10 folds. The evaluation metric is ROC-AUC score.

**Feature Extraction Setting**: the setting is almost the same as transfer learning. Except that we fix the pre-trained GNN, use it as feature extractor to get graph representations of all the data in downstream tasks, and then train a linear classifiers on top.

## 4.2 BASELINES AND MODEL CONFIGURATION

We consider five baselines, including non-pretrain (direct supervised learning) and four state-of-the-art GNN self-supervised learning (SSL) methods.

**InfoGraph** (Sun et al., 2019) maximizes the mutual information between the representations of the whole graphs and the representations of its substructures at different granularity.

**Context prediction** (Hu et al., 2020b) predicts surrounding graph structures of each node, so nodes appearing in similar structural contexts will be mapped to nearby representations.

**GPT-GNN** (Hu et al., 2020c) predicts masked edges and masked node attributes. The edge prediction makes node representations to be close when there are edges between them. The attribute prediction captures how node attributes are distributed over all graphs.

**GROVER** (Rong et al., 2020) first uses professional software, e.g. RDKit(Landrum et al., 2006), to extract functional groups (motifs) from a whole dataset. Using these motifs as a label set, each molecule is assigned a label representing which motif shows up in it and which doesn't. The model is then pre-trained by predicting this motif label as a multi-class classificaiton problem.

The state-of-the-art GNN model, Deeper Graph Convolutional Networks (DeeperGCNs) proposed in Li et al. (2020), is used as the base GNN encoder for *MICRO-Graph* and all baselines. We use the same hyperparameters for all experiments. Details about hyperparameters and model configurations are in Appendix F.

## 4.3 EVALUATION RESULT

The evaluation results under transfer learning setting and feature extraction setting is illustrated in Table 1 and Table 2. For both setting, the proposed *MICRO-Graph* outperforms all baselines on average performance and achieves the highest results on most datasets. For transfer learning setting, we gain about 2.0% performance enhancement against non-pretrain baseline. This shows the effectiveness of our self-supervised learning framework for pre-training GNNs.

| SSL methods | bace | bbbp | clintox | hiv | sider | tox21 | toxcast | Average |
|---|---|---|---|---|---|---|---|---|
| Non-Pretrain | $72.80 \pm 2.12$ | $82.13 \pm 1.69$ | $74.98 \pm 3.59$ | $73.38 \pm 0.92$ | $55.65 \pm 1.35$ | $76.10 \pm 0.58$ | $63.34 \pm 0.75$ | 71.19 |
| ContextPred | $73.02 \pm 2.59$ | $80.94 \pm 2.55$ | $74.57 \pm 3.05$ | $73.85 \pm 1.38$ | $54.15 \pm 1.54$ | $74.85 \pm 1.28$ | $63.19 \pm 0.94$ | 70.65 ( -0.54) |
| InfoGraph | $76.09 \pm 1.63$ | $80.38 \pm 1.19$ | $\mathbf{78.36 \pm 4.04}$ | $72.59 \pm 0.97$ | $56.88 \pm 1.80$ | $76.12 \pm 1.11$ | $64.40 \pm 0.84$ | 72.11 (+0.93) |
| GPT-GNN | $75.56 \pm 2.49$ | $83.35 \pm 1.70$ | $74.84 \pm 3.45$ | $74.82 \pm 0.99$ | $55.59 \pm 1.58$ | $76.34 \pm 0.68$ | $64.76 \pm 0.62$ | 72.18 (+0.99) |
| GROVER | $75.22 \pm 2.26$ | $83.16 \pm 1.44$ | $76.8 \pm 3.29$ | $74.46 \pm 1.06$ | $56.63 \pm 1.54$ | $\mathbf{76.77 \pm 0.81}$ | $64.43 \pm 0.8$ | 72.5 (+1.31) |
| MICRO-Graph | $\mathbf{76.16 \pm 2.51}$ | $\mathbf{83.78 \pm 1.77}$ | $77.50 \pm 3.35$ | $\mathbf{75.51 \pm 0.67}$ | $\mathbf{57.28 \pm 1.09}$ | $76.68 \pm 0.36$ | $\mathbf{65.42 \pm 0.62}$ | $\mathbf{73.19\ (+2.0)}$ |

Table 1: Transfer learning performance (ROC-AUC) of *MICRO-Graph* compared with other self-supervised learning (SSL) baselines on molecule property prediction benchmarks. Pre-train GNNs on ogbg-molhiv dataset, fine-tune the pre-trained model on each downstream task for 10 times.

| SSL methods | bace | bbbp | clintox | hiv | sider | tox21 | toxcast | Average |
|---|---|---|---|---|---|---|---|---|
| ContextPred | $53.09 \pm 0.84$ | $55.51 \pm 0.08$ | $40.73 \pm 0.02$ | $53.31 \pm 0.15$ | $52.28 \pm 0.08$ | $35.31 \pm 0.25$ | $47.06 \pm 0.06$ | 48.18 |
| InfoGraph | $66.06 \pm 0.82$ | $75.34 \pm 0.51$ | $\mathbf{75.71 \pm 0.53}$ | $61.45 \pm 0.74$ | $54.7 \pm 0.24$ | $63.95 \pm 0.24$ | $52.69 \pm 0.07$ | 64.27 |
| GPT-GNN | $59.43 \pm 0.66$ | $71.58 \pm 0.54$ | $62.78 \pm 0.58$ | $64.08 \pm 0.36$ | $54.67 \pm 0.16$ | $68.2 \pm 0.14$ | $57.06 \pm 0.13$ | 62.53 |
| GROVER | $65.67 \pm 0.38$ | $78.47 \pm 0.36$ | $53.19 \pm 0.68$ | $69.03 \pm 0.23$ | $54.94 \pm 0.12$ | $67.63 \pm 0.13$ | $57.28 \pm 0.05$ | 63.74 |
| MICRO-Graph | $\mathbf{69.54 \pm 0.39}$ | $\mathbf{81.07 \pm 0.42}$ | $63.69 \pm 0.56$ | $\mathbf{72.74 \pm 0.15}$ | $\mathbf{55.39 \pm 0.26}$ | $\mathbf{72.91 \pm 0.12}$ | $\mathbf{61.04 \pm 0.07}$ | $\mathbf{68.05}$ |

Table 2: Feature extraction performance (ROC-AUC) of *MICRO-Graph* compared with other self-supervised learning (SSL) baselines on molecule property prediction benchmarks. Use pre-trained models to extract graph representations for each data and train linear classifiers on top. Run each experiment 5 times.

## 4.4 ABLATION STUDY

We conduct a series of ablation studies to systematically analyze how the motif learning can benefit the contrastive learning.

### 4.4.1 WHETHER MOTIF IS HELPFUL FOR SUBGRAPH SAMPLING?

As previously discussed, the main difference of our contrastive framework with existing works is that we leverage the graph-to-subgraph views for contrastive learning. We first study whether our proposed motif-guided subgraph segmenter can indeed help contrastive learning. We implement two more widely adopted heuristic subgraph sampling baselines: random walk (RW) and K-hop neighbours (K-hop). We replace our Motif-guided Segmenter (MS) component with these two heuristic sampling algorithm for the transfer learning experiments. All the other settings stay the same.

| Sampler | bace | bbbp | clintox | hiv | sider | tox21 | toxcast | Average |
|---|---|---|---|---|---|---|---|---|
| RW | 73.61 ± 2.53 | 82.24 ± 1.99 | 75.63 ± 2.86 | 73.06 ± 1.29 | 55.88 ± 1.69 | 76.14 ± 0.56 | 63.44 ± 0.76 | 71.42 (+0.23) |
| K-hop | 73.24 ± 2.65 | 82.65 ± 1.78 | 76.76 ± 3.88 | 73.48 ± 1.41 | 55.67 ± 1.51 | 76.01 ± 0.69 | 63.34 ± 0.94 | 71.59 (+0.4) |
| MSS | **76.16 ± 2.51** | **83.78 ± 1.77** | **77.50 ± 3.35** | **75.51 ± 0.67** | **57.28 ± 1.09** | **76.68 ± 0.36** | **65.42 ± 0.62** | **73.19 (+2.0)** |

Table 3: Ablation study: analyzing the influence of subgraph sampler.

| # Motif | bace | bbbp | clintox | hiv | sider | tox21 | toxcast | Average |
|---|---|---|---|---|---|---|---|---|
| 5 | 75.81 ± 2.38 | 82.65 ± 1.97 | 76.95 ± 2.44 | 74.53 ± 1.12 | 56.78 ± 1.64 | **77.01 ± 0.89** | 64.45 ± 0.59 | 72.59 (+1.4) |
| 20 | **76.16 ± 2.51** | **83.78 ± 1.77** | **77.50 ± 3.35** | **75.51 ± 0.67** | 57.28 ± 1.09 | 76.68 ± 0.36 | **65.42 ± 0.62** | **73.19 (+2.0)** |
| 100 | 75.98 ± 2.36 | 83.68 ± 1.62 | 76.91 ± 2.57 | 75.06 ± 0.92 | **57.32 ± 1.12** | 76.98 ± 0.55 | 65.39 ± 0.69 | 73.04 (+1.85) |

Table 4: Ablation study: analyzing the influence of different motif numbers.

As shown in Table 3, replacing MS with heuristic subgraph samplers will significantly influence the performance. With random walk or k-hop sampling, graph-to-subgraph contrastive learning can only bring in 0.2-0.4% average performance enhancement against non-pretrain, which are far less than MS. This shows that the key for the overall performance enhancement of *MICRO-Graph* is not only the graph-to-subgraph views, but also the informative motif-guided subgraphs. For the details of each sampling strategy and the corresponding subgraphs examples, please refer to Appendix D.

### 4.4.2 WHETHER MOTIF NUMBER WILL INFLUENCE CONTRASTIVE LEARNING?

Number of motif slots, $K$, is an important hyperparameter in our motif learning framework. We thus conduct ablation study with three different $K$ values, 5, 20, and 100. As illustrated in Table 4, with different $K$ values, MICRO-Graph can consistently enhance the transfer performance by a large margin. Among the three numbers used, the middle one (20) gives the best result on average.

### 4.5 VISUALIZATION OF THE LEARNED MOTIFS

We further show learned motifs by collecting the closest subgraphs to them. As illustrated in Figure 3, *MICRO-Graph* automatically learns motifs that are similar to meaningful functional groups in molecule domain, such as Benzene rings and acetate. This shows that *MICRO-Graph* can learn reasonable and meaningful motifs. A complete list of the learned motifs is shown in Appendix C.

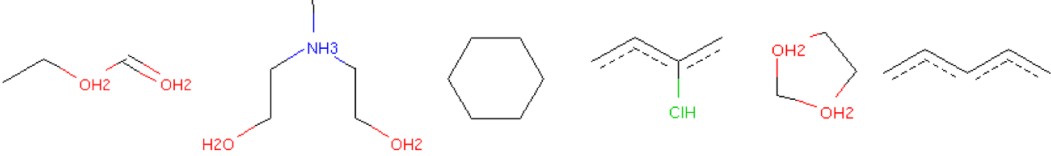

Figure 3: (Figure updated to higher resolution) Top-6 frequently occurred motifs, represented by their closest subgraph.

## 5 CONCLUSION

In this paper, we propose *MICRO-Graph* to pre-train a GNN in a self-supervised manner to automatically extract graph motifs from large-scale graph datasets. In addition, the learned motifs can guide the generation of more informative subgraphs, and help to conduct graph-to-subgraph contrastive learning. The motif learning and contrastive learning are mutually reinforced, and eventually help pre-train a generalizable GNN encoder. By pre-training on ogbg-molhiv molecule dataset with *MICRO-Graph*, we can learn meaningful motifs that align with existing molecular functional groups. Meanwhile, fine-tune the pre-trained GNN on seven chemical property prediction benchmarks yielding 2.0% average improvement over non-pretrained GNNs and outperforming other self-supervised pre-training baselines.

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

## A  PSEUDOCODE OF THE MAIN ALGORITHM

```
1   # temperature parameters: tau_g, tau_n
2   # weight parameters: lamb_m, lamb_c, lamb_s
3
4   model = Motif(args) # model contains all motif vectors, model.motifs
5   encoder = GNN(args) # GNN encoder for pretraining
6
7   for G in loader:
8       # sample subgraphs from each whole graph and return the node
9       # indices of these subgraphs => I
10      # return how many subgraphs have been sampled from each
11      # whole graph => num_subs
12      # compute the node-node affinity matrix A,
13      # return the sum of affinity of nodes within each subgraph => sum_A
14      # I: list of len N, num_subs: M x 1, sum_A: N x 1
15      I, num_subs, sum_A = segmenter(G, encoder)
16
17      # encode the input graph to get node embeddings
18      n = encoder(G)
19
20      # pool all node embeddings together for the whole graph embedding
21      h = aggregate(n, G) # M x D
22
23      # pool nodes belong to the subgraph for the subgraph embedding
24      e = aggregate(n, G, I) # N x D
25
26      # compute motif-subgraph similarity
27      S = pairwise_cosine_sim(model.motifs, e) # K x N
28
29      with torch.no_grad():
30          # compute similarity threshold eta as the top 10% for each motif
31          S_top10, _ = S.topk(k=int(0.1 * S.shape[1]), dim=1) # K x 0.1*N
32          eta = S_top10[:, -1] # K x 1
33          Q = sinkhorn(S) # use the Sinkhorn-Knopp algorithm to solve for Q, K x N
34
35      # identy whether a subgraph is similar enough to a motif
36      S_mask = (S > eta).sum(dim=0) > 0 # 1 x N
37
38      # compute the sampler loss
39      loss_s = sum_A[S_mask]
40
41      # normalize S
42      S_tilde = torch.softmax(S / tau_g, dim=0) # K x N
43
44      # compute motif learning loss
45      loss_m = - (Q * S_tilde.log()).sum(dim=0).mean()
46
47      # compute pairwise similarities between each
48      # whole graph embeddings and subgraph embeddings
49      W = torch.softmax(pairwise_cosine_sim(h, e)/ tau_g, dim=1)
50
51      # compute the contrastive loss
52      blocks = [torch.ones(1, int(n)) for n in num_subs]
53      W_mask = torch.block_diag(*blocks)
54      loss_c = - (W_mask * W.log()).sum(dim=1).mean()
55
56      # final loss
57      loss = lamb_m*loss_m + lamb_c*loss_c + lamb_s*loss_s
58
59
60  def segmenter(G, encoder):
61      with torch.no_grad():
62          I = []
63          num_subs = []
64          sum_A = []
65          for G_i in G:
66              # encode the input graph to get node embeddings
67              n = encoder(G_i)
68
69              # compute the node-node affinity matrix A_i
70              A_i = torch.softmax(pairwise_cosine_sim(n, n) / tau_n, dim=1)
71
72              # apply spectral clustering and find connected components
73              # to segment subgraphs, I_i is a list
74              I_i = find_connected_components(spectral_clustering(A_i))
75
76              # the node indices of these subgraphs
77              I += I_i
78
```

```
79                  # how many subgraphs are sampled from each whole graph
80                  num_subs += [len(I_i)]
81
82                  # the sum of affinity values of nodes within each subgraph
83                  sum_A += [(A_i[index][:, index]).sum() for index in I_i]
84
85          return I, num_subs, sum_A
86
87
88  def sinkhorn(S, num_iters=3, lamb=20):
89      '''
90      Implementation of the sinkhorn function adopted from
91      https://github.com/facebookresearch/swav/blob/master/main_swav.py
92      '''
93      with torch.no_grad():
94          Q = torch.exp(S).t()
95          Q /= torch.sum(Q)
96          u = torch.zeros(Q.shape[0]).to(Q.device)
97          r = torch.ones(Q.shape[0]).to(Q.device) / Q.shape[0]
98          c = torch.ones(Q.shape[1]).to(Q.device) / Q.shape[1]
99
100         curr_sum = torch.sum(Q, dim=1)
101         for it in range(num_iters):
102             u = curr_sum
103             Q *= (r / u).unsqueeze(1)
104             Q *= (c / torch.sum(Q, dim=0)).unsqueeze(0)
105             curr_sum = torch.sum(Q, dim=1)
106         return (Q / torch.sum(Q, dim=0, keepdim=True)).t().float()
```

## B    NOTATION SUMMARY

Below, we summary the important notations and symbols used paper in the order they appeared.

Graph level
| | |
|---|---|
| $M$ | Total number of whole graphs |
| $\mathcal{G}_i$ | Whole graphs |
| $\boldsymbol{h}_i$ | Whole graph embeddings |

Subgraph level
| | |
|---|---|
| $N$ | Total number of subgraphs |
| $g_j$ | Subgraphs |
| $\boldsymbol{e}_j$ | Subgraph embeddings |

Node level
| | |
|---|---|
| $\boldsymbol{n}_i$ | Node embedding |
| $\boldsymbol{A}^{(i)}$ | Node affinity matrix, n-by-n dimensional for a whole graph with n nodes |
| $\boldsymbol{A}^{(i)}_{s,t}$ | Node affinity between node s and node t in a graph |
| $\mathbb{I}$ | Indices of nodes forming a subgraph selected by the *Motif-guided Segmenter* |

Motifs
| | |
|---|---|
| $K$ | Number of motifs |
| $\boldsymbol{m}, \boldsymbol{m}_k$ | Motif vectors |
| $\boldsymbol{S}$ | Similarities between all K motifs and all N subgraphs, K-by-N dimensional |
| $\boldsymbol{s}_j$ | Similarities between all K motifs and the subgraph j, K-by-1 dimensional |
| $S_{k,j}$ | Similarity between the motif k and the subgraph j, scalar |
| $\tilde{S}_{k,j}$ | Normalized similarity between the motif k and the subgraph j |
| $\boldsymbol{Q}$ | Motif-based cluster assignment matrix, K-by-N dimensional |
| $\boldsymbol{q}_j$ | Motif-based cluster assignment of subgraph j, 1-by-N dimensional |
| $\eta_k$ | Threshold for deciding whether a subgraph is similar enough to motif k |

Contrastive
| | |
|---|---|
| $\boldsymbol{W}$ | Normalized similarities between M graphs and N subgraphs, M-by-N dimensional |
| $W_{i,j}$ | Normalized similarities between graph i and subgraph j |

Others
| | |
|---|---|
| $\tau_n$ | Temperature for the softmax normalization of node-node similarity |
| $\tau_g$ | Temperature for the softmax of motif-subgraph and graph-subgraph similarity |
| $\boldsymbol{E}(\cdot)$ | GNN encoder used to generate node embeddings and (sub)graph embeddings |
| $D$ | Dimension of node embeddings, (sub)graph embeddings, and motif vectors |

## C    TOP K CLOSESET SUBGRAPHS TO LEARNED MOTIFS

Examples of the first 10 learned motifs of the ogbg-molhiv dataset is shown in Figure 4 and 5.

Figure 4: (Figure updated to higher resolution) Motif 1-5, represented by top k closest subgraphs to the learned motif representations. Each row represents a motif, represented by some subgraphs that is closest to these motifs. Three columns indicates top 1, top 2, and top 3 most similar subgraph respectively.

Figure 5: (Figure updated to higher resolution) Motif 6-10, represented by top k closest subgraphs to the learned motif representations. Each row represents a motif, represented by some subgraphs that is closest to these motifs. Three columns indicates top 1, top 2, and top 3 most similar subgraph respectively.

## D    SAMPLING STRATEGIES AND SAMPLED SUBGRAPHS

Here we describe the details of our heuristic sampling strategies.

For random walk, we use a random walk length uniform in [10, 40]. Starting from a randomly selected seed node, we randomly select its neighborhood as next hop, until reaching the walk length threshold.

For $K$-hop neighbors, we pick hop number $k$ to be 1 or 2 with equal probability. Starting randomly selected seed node, we collect all the neighbors within $k$ hop as the sampled subgraph.

We also shows some subgraph examples generated by these two heuristic strategies and our proposed motif-guided subgraph segmenter in Figure 6. From the sampled subgraphs, we can see that random walk is more likely to generate chains, while k-hop sampling is more likely to generate half part of a Benzene ring. Neither of these two heuristic approaches can successfully generate a complete and clean functional group, and the generated subgraphs are not that meaningful. On the contrary, our motif-guided sampler can succssfully generate a complete benzene ring and two other molecule substructures. This intuitively explains why the graph-to-subgraph contrastive learning can only work with our proposed subgraph segmenter.

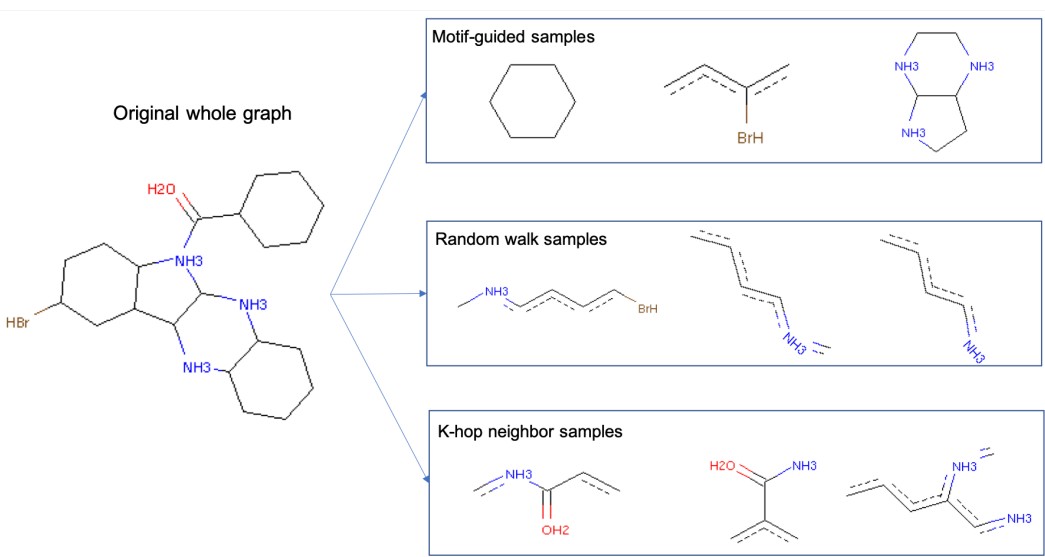

Figure 6: (Figure updated to higher resolution) Comparison between different sampling strategies. The original graph is shown on the left. Samples produced by three different sampling strategies are shown on the right. The top row shows the samples by our motif-guided segmenter. The following two rows corresponding to random walk samples and k-hop samples.

## E    CHEMICAL PROPERTY PREDICTION BENCHMARKS

In our experiments, we evaluated model performance on seven Open Graph Benchmark (OGB) molecule property prediction datasets. We provide a synopsis of each downstream task dataset from Hu et al. (2020b) below:

- **bace:** Qualitative binding results for a set of inhibitors of human $\beta$-secretase 1.
- **bbbp:** Blood-brain barrier penetration (membrane permeability).
- **clintox:** Qualitative data classifying drugs approved by the FDA and those that have failed clinical trials for toxicity reasons.
- **hiv:** Experimentally measured abilities to inhibit HIV replication.

| Dataset | bace | bbbp | clintox | hiv | sider | tox21 | toxcast |
|---------|------|------|---------|-----|-------|-------|---------|
| # graphs | 1513 | 2039 | 1477 | 41127 | 1427 | 7831 | 8576 |
| # nodes | 51577 | 49068 | 38637 | 1049163 | 48006 | 145459 | 161088 |
| # edges | 111536 | 105842 | 82372 | 2259376 | 100912 | 302190 | 161088 |
| # tasks | 1 | 1 | 2 | 1 | 27 | 12 | 617 |

Table 5: Statistics on number of graphs, nodes, edges, and tasks in each OGB molecule dataset.

- **sider:** Database of marketed drugs and adverse drug reactions (ADR), grouped into 27 system organ classes.
- **tox21:** Toxicity data on 12 biological targets, including nuclear receptors and stress response pathways.
- **toxcast:** Toxicology measurements based on over 600 in vitro high-throughput screenings.

Table 5 summarizes important statistics of the OGB molecule datasets related to the number of graphs, the size of graphs, and number of properties that require prediction for each molecule. For these datasets, there are 9-dimensional node features including atomic number, chirality, and etc. There are also 3-dimensional edge features including bond type, bond stereochemistry, and an additional bond feature indicating whether the bond is conjugated. For further information on the OGB datasets, please refer to Hu et al. (2020b) and Hu et al. (2020a).

## F  HYPERPARAMETERS AND MODEL CONFIGURATION

We show the hyper-parameters we used for running our experiments. We use the same hyperparameters, 5 hidden layers and 300 hidden dimension, recommended in Li et al. (2020) for all DeeperGCN models. We pre-train our model using Adam optimizer for 100 epochs, with batch size (number of graphs per batch) 512. For fine-tuning, we train the model for 100 epochs with batch size 32. We select model with highest validation result and report its test result. Corresponding experiment results are shown in Section 4.3.

The parameters for running context prediction baseline is shown in Table 6. We show additional experiments of ContextPred with different parameters in Table 7.

| Context represent mode | cbow |
|------------------------|------|
| Context pooling | mean |
| Negative sample ratio | 1 |
| Context size | 3 |
| r1 | 4 |
| r2 | 7 |

Table 6: Hyper-parameters of the context prediction pretraining

| | bace | bbbp | clintox | hiv | sider | tox21 | toxcast | Average |
|---|------|------|---------|-----|-------|-------|---------|---------|
| r1=2, r2=5 | $73.55 \pm 2.5$ | $81.7 \pm 2.84$ | $75.82 \pm 4.11$ | $73.79 \pm 0.9$ | $54.98 \pm 1.48$ | $75.6 \pm 0.78$ | $63.88 \pm 0.76$ | 71.33 |
| r1=3, r2=4 | $72.65 \pm 2.32$ | $81.23 \pm 1.98$ | $73.14 \pm 6.8$ | $73.67 \pm 0.99$ | $53.99 \pm 1.38$ | $75.67 \pm 0.65$ | $63.53 \pm 0.79$ | 70.55 |

Table 7: Additional experiments for ContextPred with different parameters r. Note: results shown in Section 4 are with r1 = 4 and r2 = 7

## G  MOTIF CLUSTER SIZE DISTRIBUTION

In Figure 7, we show the distribution of cluster sizes of all the learned motifs. Although with the equal-size constraint, the distribution is not completely uniform.

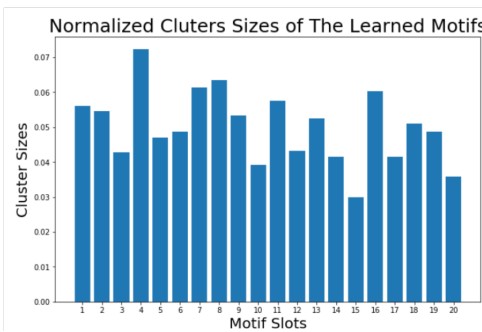

Figure 7: Distribution of cluster sizes of all the learned motifs

## H    VISUALIZING SIMILARITIES

In this section, we visualize the similarity scores between the learned graph and subgraph representations. In particular, we consider a whole graph $\mathcal{G}_1$ within a batch of graphs $\{\mathcal{G}_1, ..., \mathcal{G}_B\}$, and three different subgraphs $g_1$, $g_2$, and $g_3$ sampled from $\mathcal{G}_1$.

In Figure 8, we visualize the distribution of pairwise cosine similarity between all graph-motif-assignment vectors $a_i$. Here each $a_i$ is associated with a graph $\mathcal{G}_i$ with $N_i$ subgraphs $g_i, \ldots, g_{N_i}$. It is a K-dimensional vector representing which motif the graph $\mathcal{G}_i$ contains. We compute $a_i$ by aggregating the corresponding normalized motif-subgraph similarities $\tilde{S}$ as the following.

$$a_i = \frac{1}{|N_i|} \sum_{g_j \in \mathcal{G}_i} \tilde{S}_{:,j} \qquad (10)$$

This distribution in Figure 8 is over the pairwise cosine similarities of all $a_i$'s in the batch. In this case, only 8.7% of these pairwise cosine similarities are higher than 0.5, and only 1.7% higher than 0.9, which shows the graph dataset is well distributed and contains diverse whole graphs. It is relatively uncommon for whole graphs to share many subgraphs.

In Figure 9, we visualize the distribution of similarity scores between $G_1$ and all the subgraphs sampled from the whole batch of graphs $\{\mathcal{G}_1, ..., \mathcal{G}_B\}$. We see that the distribution is centered around 0, with maximum roughly equal to 0.6.

In Figure 10, we visualize similarity scores between $\mathcal{G}_1$ and $g_1$, $g_2$, and $g_3$, and we zoom in to each dimension. The similarity score we use is cosine similarity. In this case, the cosine similarity scores are 0.6026, 0.6020, and 0.4786 respectively, which are significantly higher than subgraphs sampled from other whole graphs as shown in Figure 9. To figure out how we got these high scores, we can zoom into each dimension. In other words, we check the elementwise product of the 300-dimensional graph and subgraph representation vectors, without summing these 300 number together. We find that the three distribution corresponding to these subgraphs look very different, which indicates these three subgraph representations activate different dimensions of the multi-view whole graph representation. In other words, they are only similar to the projection of the whole graph representation on different basis.

In Figure 11, we further show that pairwise similarity scores between these three subgraphs $g_1$, $g_2$, and $g_3$, which are not very high. This verifies our claim in Figure 10.

In Figure 12, we show the pairwise similarity scores between the first 30 subgraphs sampled from $\{\mathcal{G}_1, ..., \mathcal{G}_B\}$. We see that even though these subgraphs are listed in order, i.e. $g_1, ..., g_3$ are from $\mathcal{G}_1, g_4, g_5$ are from $\mathcal{G}_2, g_6, ...g_8$ are from $\mathcal{G}_3$, and etc, similarity scores are roughly uniform. In other words, this heat matrix is not strictly block diagonal, indicating subgraphs from the same whole graph do not necessarily have high similarities among them.

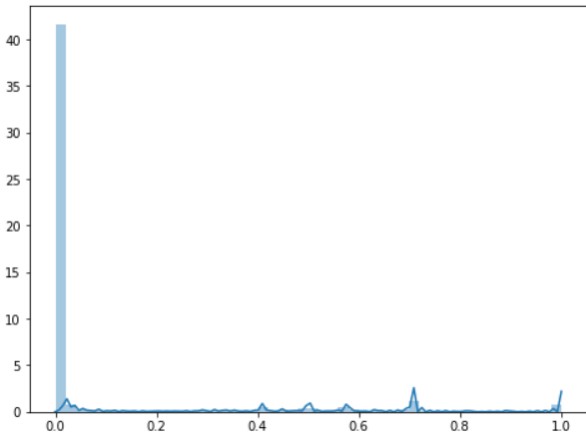

Figure 8: Pairwise similarity scores of all graph-motif-assignment vectors $a_1, ..., a_B$

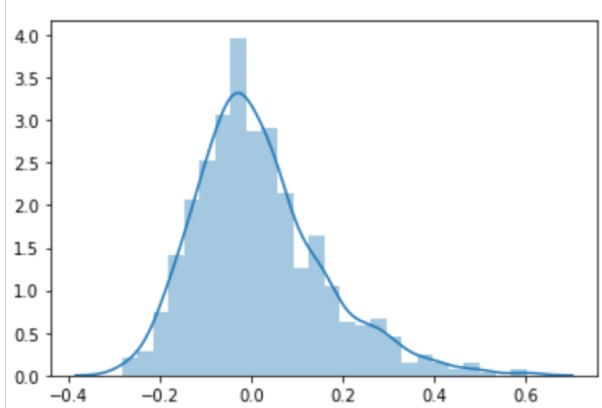

Figure 9: Distribution of similarity scores between the whole graph $G$ and all the subgraphs

## I  LINEAR EVALUATION IN ACCURACY

We show the linear evaluation result in **accuracy** of our model and the baselines in Table **??**. Note that the results of tables in Section 4 are in AUC-ROC rather than accuracy.

| SSL methods | bace | bbbp | clintox | hiv | sider | tox21 | toxcast | Average |
|---|---|---|---|---|---|---|---|---|
| ContextPred | $57.13 \pm 0.55$ | $80.58 \pm 0.21$ | $\textbf{93.03} \pm \textbf{0.00}$ | $\textbf{96.49} \pm \textbf{0.00}$ | $75.35 \pm 0.08$ | $92.21 \pm 0.01$ | $83.68 \pm 0.02$ | 82.64 |
| InfoGraph | $67.51 \pm 1.83$ | $82.08 \pm 1.13$ | $90.92 \pm 0.40$ | $93.24 \pm 1.92$ | $68.80 \pm 0.52$ | $89.60 \pm 0.37$ | $80.19 \pm 0.06$ | 81.76 |
| GPT-GNN | $59.33 \pm 0.26$ | $73.82 \pm 2.06$ | $93.01 \pm 0.07$ | $94.16 \pm 3.06$ | $70.86 \pm 0.34$ | $88.60 \pm 0.26$ | $81.20 \pm 0.12$ | 80.14 |
| MICRO-Graph | $\textbf{76.49} \pm \textbf{0.25}$ | $\textbf{85.44} \pm \textbf{0.20}$ | $93.01 \pm 0.05$ | $94.49 \pm 0.00$ | $\textbf{75.82} \pm \textbf{0.00}$ | $\textbf{92.74} \pm \textbf{0.00}$ | $\textbf{84.24} \pm \textbf{0.05}$ | **86.32** |

Table 8: Feature extraction performance (ACC) of MICRO-Graph compared with other self-supervised learning (SSL) baselines on molecule property prediction benchmarks. Use pre-trainedmodels to extract graph representations for each data and train linear classifiers on top. Run eachexperiment 5 times

## J  A SYNTHETIC DATASET TO STUDY MOTIF LEARNING

In addition to studying the pre-training in chemical domain, we also construct a synthetic dataset to that align with our assumptions to verify the effectiveness of the propose method. We assume there exist $K$ graph motifs, and each whole graph can be represented by certain combinations of

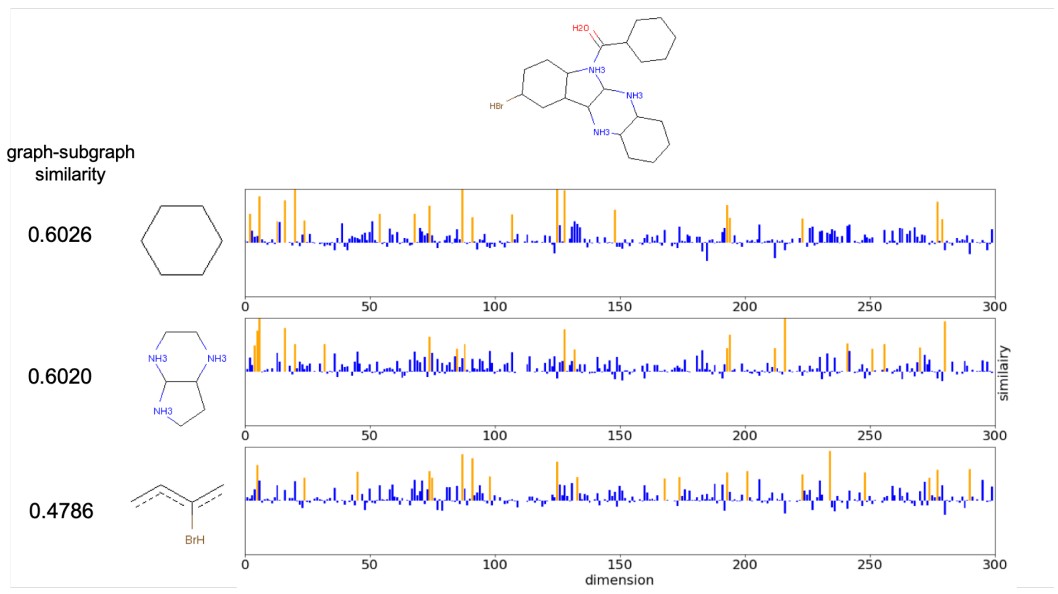

Figure 10: Similarity between the whole graph $\mathcal{G}_1$ and three subgraphs $g_1$, $g_2$, and $g_3$, zoom in to each dimension. For each row, x-axis is the dimension slot 1 to 300, and y-axis is the similarity scores between corresponding dimensions of the whole graph representation and each subgraph representation. We indicate the top 20 scores in orange. We can see that these three subgraphs have very different similarity score distributions, though summing over all 300 dimensions give alike high scores.

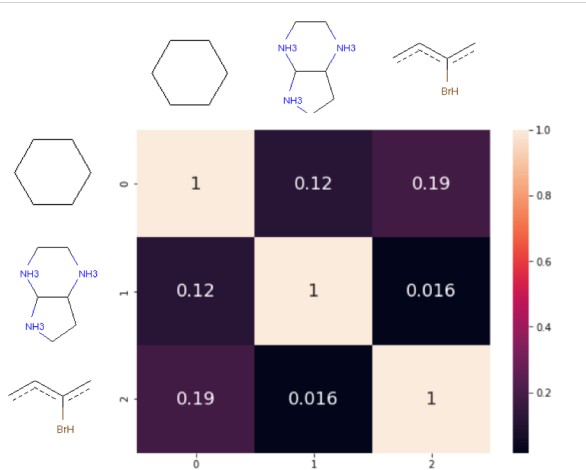

Figure 11: Pairwise similarity scores between subgraphs $g_1$, $g_2$, and $g_3$

these motifs. Following this rule, we first select some graph structures as motifs, and randomly sample some combinations, and generate graphs (as is illustrated in Figure 13) by combining the motifs, with some randomly added or deleted nodes and edges. Each graph will also be assigned a corresponding one-hot vector label, whose dimension is the total number of combinations we generated for the dataset. Eight subgraph templates and examples of generated whole graphs are shown below. As is illustrated in Figure 14, our *MICRO-Graph* can successfully learn the underlying graph motifs and templates without any annotations.

The advantage of constructing such synthetic dataset is that we can know the underlying ground-truth of graph motifs and combination rules. We believe on top of this toy dataset, more complex

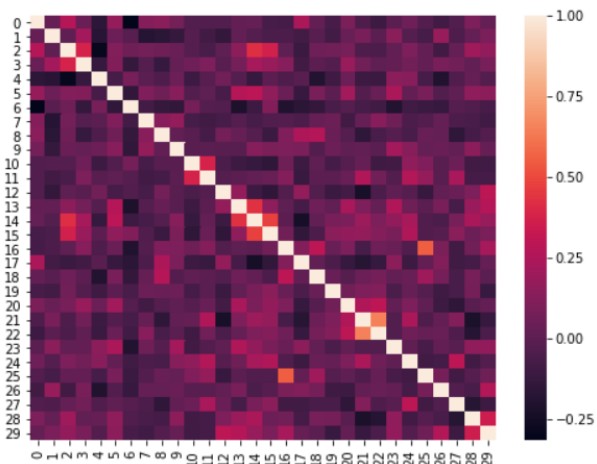

Figure 12: Pairwise similarity scores between the first 30 subgraphs sampled from $\{\mathcal{G}_1, ..., \mathcal{G}_B\}$

syntax and semantic knowledge can be incorporated for automatic motif and even rule mining communities.

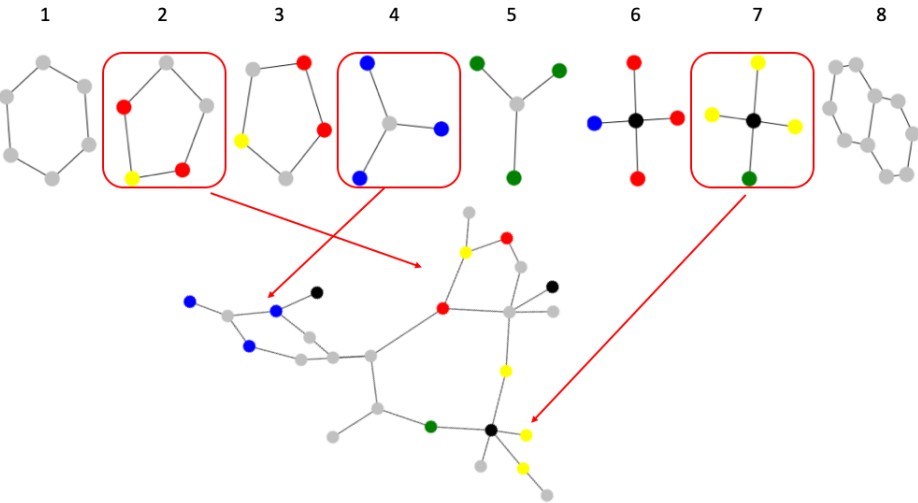

Figure 13: Example of a synthetic graph. The upper 8 graphs are the base motifs, and we generate this graph with a combination [2, 4, 7]. Different colors represent different node features.

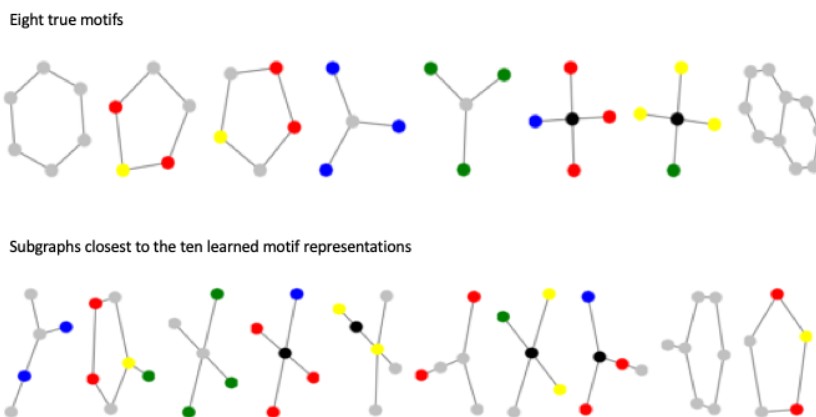

Figure 14: Comparison of the eight motifs used to generate the synthetic dataset and subgraphs closest to the ten learned motif representations

