# OpenReview forum: "Motif-Driven Contrastive Learning of Graph Representations"
_ICLR.cc/2021/Conference — Reject_

### Official Review · AnonReviewer2 · 2020-10-27
**useful idea but with confusing notations and abnormal experimental results**

**Rating:** 5
**Confidence:** 5

**Review:**

This paper proposes to learn the sub-graph patterns from a collection of training graphs. The key idea is to partition each graph into segments and enforce a global clustering of the subgraphs. The partitioning is also guided through contrastive learning, i.e., subgraphs should have a larger similarity with the graph it is drawn from, compared with other graphs. The learned GNN (that generates node embedding) will then be used to some downstream learning tasks with or without further fine-tuning.

The notation of the paper is very hard to follow. For example, no where is N defined, which I assume is the number of sug-graphs. Capital latters S_i’s represent vector (not sure if my guess is correct), as lower-case letter q_j’s, which somewhat affects the reading. Also the definition of n_i, and n_{i,s} is not quite following the custom of matrices. Furthermore, what is the relation between s_i and S_i (both board), and what is the difference between S (bold) and S (not bold)? The h_i is defined but not used, and in equation (7) g_i is used, which I assume is the summary embedding vector for graph G_i; in (7) s_i seems to be subgraph embedding vector, and this makes me more confused about the s_{j,k} appearing in the beginning of section 3.2 which is defined as the cosine similarity between subgraph j and motif k. The E[G_i] is with bold E somewhere and non-bold letter E elsewhere. Altogether, there are S (bold), S(not bold), s_{j,k} (bold) that seems to be a similarity measure, and s_i (bold) which seems to represent subgraph vector. It’s quite confusing to me. Can authors clearly define these symbols when they are used, and make sure they are consistent throughout the paper, and explicitly mark their dimensionality to avoid confusion? Usually upper-case letters are for matrices and lower-case bold letters for vectors.

The idea of partitioning graphs into sub-graphs are a useful idea in breaking the complexity of graph-structured objects and exploring the potential hierarchical organizations of the graph. Using contrastive learning as a self-supervision may further improve the partitioning of each graph. However, the grouping part and the contrastive part may conflict with each other in that some sub-graphs are shared among different graphs, which can be quite common in chemical compounds.
Under (5), it is mentioned that spectral clustering is used to partition the graphs; is it done end-to-end and if so where is the loss function corresponding to this operation?

In (6), (s,t) in g(i,j): What is g(i,j) in particular? a threshold eta is used in the indicator function and how to choose eta (considering that it is used directly on a set of variables)? Is (6) end-to-end optimizable?

In Figure~2, what is meant by the blue and red markers (like + and -)?

Experimental results are quite strange in that the transfer learning setting (which further finetunes the learned GNN based on a small set of labels, as in Table 1) leads to even worse performance than the feature extraction setting (in which no fine-tuning is performance, as shown in Table2) and the gap can be as huge as 15% in accuracy!

---

> ### Author Response · Authors · 2020-11-14
> **Question answering continued**
>
> Q: Spectral clustering for partition graphs and the segmenter loss (equation (6))
>
> A: The spectral clustering for partition graphs is not end-to-end. It is only used for grouping nodes into subgraphs, and the subgraph assignment is only served as “labels” for the loss function in equation (6) to encourage nodes within the same cluster to have closer embeddings.
>
> Specifically, Given a graph $G_i$ with n nodes, we first constructs a n-by-n node-node affinity matrix $A^{(i)}$, and then apply spectral clustering to this matrix $A^{(i)}$, as if $A^{(i)}$ is a n-vertices complete graph with affinity scores as edge weights. Then we are done with spectral clustering.
>
> The segmenter loss term $L_s$ in equation (6) is meant to make nodes belong to a subgraph (predicted by spectral clustering) to have similar node embeddings, if this subgraph is similar to a learned motif.
>
> Q: How to choose $\eta$?
>
> $\eta$ is the threshold for deciding whether a subgraph is similar enough to a learned motif. It is dynamically set to take the top 10% most similar subgraphs to each motif. We hope this part is clear in the updated paper version, please refer to the pseudocode in Appendix A as well.
>
>
> Q: Blue and red markers in Figure 2
>
> A: For contrastive learning, we need to define positive pairs and negative pairs. In Figure 2, when two arrows are pointing a red plus sign, the two objects these arrows are referring to are considered as positive pairs. Vice versa for the negative pairs.
>
> Q: Experimental results
>
> A: We are sorry for the confusion caused by the fine tune evaluation and linear evaluation. The original Table1 and Table2 in the paper were not directly comparable, as they were meant to compare with two previous works [1] and [2]. Therefore, we closely followed the settings of these two works respectively, where the evaluation metric of Table1 was AUC-ROC score and the evaluation score of Table 2 was accuracy. We have now updated Table2, so both cases use AUC-ROC score. The old version of Table2 has been moved to Appendix. Below we show the updated Table 2 with AUC-ROC score.
>
>
>
>
>
>
> |SSL methods   | bace              | bbbp             |clintox           |hiv                 |sider              | tox21            |toxcast           |Avg|
> |-------------------|--------------|--------------|----------------|----------------|----------------|----------------|----------------|----------------|
> |ContextPred    | 53.09 ± 0.84 | 55.51 ± 0.08 | 40.73 ± 0.02 | 53.31 ± 0.15 | 52.28 ± 0.08 | 35.31 ± 0.25 | 47.06 ± 0.06 | 48.18 |
> |InfoGraph        | 66.06 ± 0.82 | 75.34 ± 0.51 | 75.71 ± 0.53 | 61.45 ± 0.74 | 54.7 ± 0.24   | 63.95 ± 0.24 | 52.69 ± 0.07 | 64.27 |
> |GPT-GNN         | 59.43 ± 0.66 | 71.58 ± 0.54 | 62.78 ± 0.58 | 64.08 ± 0.36 | 54.67 ± 0.16 | 68.2 ± 0.14   | 57.06 ± 0.13 | 62.53 |
> |MICRO-Graph | 69.54 ± 0.39 | 81.07 ± 0.42 | 63.69 ± 0.56 | 72.74 ± 0.15 | 55.39 ± 0.26 | 72.91 ± 0.12 | 61.04 ± 0.07 | 68.05|
>
> The updated linear evaluation result shows that our framework outperforms all three baselines. The comparison between linear evaluation and fine-tune evaluation is also consistent with our expectation (and also yours~). As you pointed out, the linear results should be lower than the fine-tune result. In particular, MICRO-Graph’s performance is 5.14% lower than fine-tune on average, and for three baselines, the average performance is lower by 22.47%, 7.84%, and 9.65% respectively. This shows that MICRO-Graph has learned more robust representations during pre-training, and thus a simple linear classifier on top of it can achieve reasonably good results.
>
>
> We hope our answer has made things clear. We are happy for any further discussions or questions, so please feel free to let us know if any of the concerns are not fully addressed.
>
> Reference
>
> [1] Weihua  Hu,  Bowen  Liu,  Joseph  Gomes,  Marinka  Zitnik,  Percy  Liang,  Vijay  Pande,  and  Jure Leskovec. Strategies  for  pre-training  graph  neural  networks. In International  Conference on  Learning  Representations,  2020b
>
> [2]Fan-Yun Sun, Jordan Hoffmann, Vikas Verma, and Jian Tang. Infograph: Unsupervised and semi- supervised graph-level representation learning via mutual information maximization, 2019.

---

> > ### Comment · AnonReviewer2 · 2020-11-23
> > **on novelty and design of sub-graphs**
> >
> > The authors have addressed some of the questions.
> >
> > Overall, the paper follows the idea of Contrastive Predictive Coding and borrow that idea to graph domain. Namely pulling together sub-structures from the same graph and pulling apart sub-structures from different graphs. The main adjustment is the identification of sub-graphs (unlike image patches that can be identified trivially). The authors used a grouping loss to encourage a global grouping structure of the sub-graphs, which seems to be the main contribution. However, segmentation may not be the best way to identify sub-graphs considering the numerous possibility to partition a graph into sub-graphs; for example, Qiu's method of contrastive graph coding describes different ways to obtain/design sub-graphs. In this work, sub-graphs are determined off-line using spectral clustering in each graph separately, which can be restrictive and  less optimal, making the novelty of the work less significant.

---

> > > ### Author Response · Authors · 2020-11-24
> > > **Clarify misunderstanding of the subgraph sampling method**
> > >
> > > Dear Reviewer2,
> > >
> > > We really appreciate your response and comments. We feel there is some misunderstanding about our paper. We hope the following response can address your concerns and clarify our method and contribution.
> > >
> > > In the following, we first clarify our segmenter in more detail. Then we discuss the advantage of our method over other structure-based-only sampling methods, e.g. methods in Qiu’s graph contrastive coding (GCC) paper. Related experiment results are shown in ablation section 4.4.1.
> > >
> > > First, our segmenter is $\textbf{not off-line}$. It is $\textbf{learnable}$ with motif guidance and it utilizes $\textbf{cross-graph information}$ for generating subgraphs. Spectral clustering is just one part of the segmenter, but not the segmenter itself. One way to see this difference is that, when the parameters in the GNN are updated, our segmenter will produce different subgraphs for the same whole graph. Also, these subgraphs will be more and more motif-like through training. In contrast, the regular off-line spectral clustering will always produce the same partition for the same whole graph, because it is structure-based-only.
> > >
> > > As we mentioned in our paper and the previous response, given a graph $G_i$ with n nodes, we first construct an n-by-n node-node affinity matrix $A^{(i)}$. The node-node affinity is computed using the node representations, which are generated by the GNN. In this case, the representation of node $j$ contains three types of information. The structure information around node $j$, the feature information of node $j$ and its (multi-step) neighbors, and more importantly, information from other whole graphs. The last is because node representations of different graphs are generated by the same GNN, and knowledge of all graphs in the dataset is captured by the GNN parameters.
> > >
> > > The spectral clustering is applied to this matrix $A^{(i)}$ rather than the graph $G_i$ by treating $A^{(i)}$ as an n-vertices complete graph with affinity scores as edge weights. Through the whole subgraph sampling process, we utilize much more information to determine which nodes should be picked as a subgraph, rather than only looking at the graph structure.
> > >
> > > The segmenter loss term $L_s$ in equation (6) improves the quality of sampled subgraphs by including motif guidance.  Specifically, if a subgraph $\boldsymbol g_i$ is similar to a learned motif, the representations of nodes belonging to $\boldsymbol g_i$ will become more similar, and vice versa. Then when we segment subgraphs in later iterations, the subgraphs will be more motif-like. In contrast, structure-based-only sampling methods won’t be able to utilize feature information, nor do they consider knowledge from other graphs. Their subgraph samples are simply determined by some initial hyper-parameters, i.e. walk length for the random walk.
> > >
> > > Common subgraph sampling methods, including random walk, k-hop neighbors, and direct spectral clustering on the graph structure, are all structure-based-only. Structure information is also what GCC is designed to learn. As Qiu said in section 3.1 of GCC, “the focus of this work is on structural representation learning without node attributes and node labels”.  In GCC, the sampling method is a random walk with restart, and the dataset has no node features, e.g. the IMDB dataset. In our case, we focus on graphs with rich features, e.g. chemical graphs, so structure-based-only methods are not optimal.
> > >
> > > In summary, we totally agree with your point that structure-based-only sampling methods are “restrictive and less optimal”. These methods don’t consider feature information, and they are single-graph-based. Our segmenter is exactly meant to solve this problem, as it considers structure information, feature information, and cross-graph information. It is shown to be effective in our ablation study 4.4.1. As shown in Table 3, the fine-tune performance of our segmenter is significantly higher than the random walk (RW) and the k-hop neighbors (K-hop).
> > >
> > >
> > > |Sampling methods   | bace              | bbbp             |clintox           |hiv                 |sider              | tox21            |toxcast           |Avg |
> > > |-------------------|--------------|--------------|----------------|----------------|----------------|----------------|----------------|----------------|
> > > RW | 73.61 ± 2.53 | 82.24 ± 1.99 | 75.63 ± 2.86 | 73.06 ± 1.29 | 55.88 ± 1.69 | 76.14 ± 0.56 | 63.44 ± 0.76 | 71.42  (+0.23)
> > > K-hop | 73.24 ± 2.65 | 82.65 ± 1.78 | 76.76 ± 3.88 | 73.48 ± 1.41 | 55.67 ± 1.51 | 76.01 ± 0.69 | 63.34 ± 0.94  | 71.59 (+0.4)
> > > Ours | 76.16 ± 2.51 | 83.78 ± 1.77 | 77.50 ± 3.35 | 75.51 ± 0.67 | 57.28 ± 1.09 | 76.68 ± 0.36 | 65.42 ± 0.62 | 73.19 (+2.0)
> > >
> > > The segmenter is an important contribution of our paper, so we hope the misunderstanding is gone by now. Please feel free to ask any more questions or make any more comments. Either about the segmenter or other parts. It’s our pleasure to have further discussions. Thanks.

---

> > > > ### Comment · AnonReviewer2 · 2020-11-24
> > > > **on spectral clustering**
> > > >
> > > > I am confused about the spectral clustering part. The authors mentioned that "A: The spectral clustering for partition graphs is not end-to-end. It is only used for grouping nodes into subgraphs, and the subgraph assignment is only served as “labels” for the loss function in equation (6) to encourage nodes within the same cluster to have closer embeddings."
> > > >
> > > > Then the authors claim that the segmentation is done end-to-end. What is meant by segmenter? is it the same as partitioning a graph/nolecule into smaller sub-graphs?
> > > >
> > > > Can you clarify that whether the partitioing process is optimized or not?  When you apply spectral clustering to partition graphs into sub-graphs, how do you incorporate "learned" feature of the nodes into consideration, if it is done off-line, and each node only has a one-hot feature vector (atom type) that is equally distant away from each other. Then it means you are only considering the structural information in the paritioning, too.

---

> > > > > ### Author Response · Authors · 2020-11-24
> > > > > **Clarifying segmenter involves spectral clustering but is not equivalent to spectral clustering**
> > > > >
> > > > > Thank you for actively providing your response and comments.
> > > > >
> > > > > We feel that your confusion comes from interpreting the segmenter and the spectral clustering as the same thing. They are not. Segmenter contains several steps, spectral clustering is one step.
> > > > >
> > > > > Yes, as you stated, our segmenter partitions a graph into smaller graphs and it is done end-to-end.
> > > > >
> > > > > Yes, the partitioning process is optimized, and we indeed incorporate learned features of nodes into consideration.
> > > > >
> > > > > The way we do that is to first use learned features to compute a node-node affinity matrix $A^{(i)}$, and then we apply spectral clustering on this matrix $A^{(i)}$. The feature information is incorporated when we compute the matrix $A^{(i)}$.
> > > > >
> > > > > Spectral clustering can be applied to any matrix, when we simply apply it to graph adjacency matrix, it only considers structural information. In our case, spectral clustering is not directly applied to the adjacency matrix, but to an affinity matrix $A^{(i)}$ that is constructed using learned features.
> > > > >
> > > > > In our previous response you cided, we mentioned the spectral clustering is not end-to-end, because spectral clustering as an algorithm itself has no loss associated with it. It only needs to do eigendecomposition. However, equation (6) is the term associated with the segmenter, because learned features are incorporated in the computation of affinity matrix $A^{(i)}$, and we optimize these features.
> > > > >
> > > > > We understand that this part is not easy to get the first time, and it is indeed a new perspective our paper contributes. To the best of our knowledge, sampling methods used in other self-supervised learning works only consider structural information.
> > > > >
> > > > > More details of how the segmenter works and why it is better can be found either in the paper or the previous response. We will keep this answer short, so the main point is clear.
> > > > >
> > > > > Please feel free to ask any more questions or make any more comments. It’s our pleasure to have further discussions. Thanks.

---

> > > > > > ### Comment · AnonReviewer2 · 2020-11-24
> > > > > > **continued**
> > > > > >
> > > > > > Spectral clustering does have a loss associated with it, like in normalized cut.
> > > > > >
> > > > > > What is confusing all the time is the way authors perform optimization. It seems that it is not a strict end-to-end optimization, but instead, authors freeze the current values of all the parameters say, after each iteration, and perform an "off-line" eigenvalue decomposition of the affinity matrix. Similar things happen in choosing the threshold parameter. The authors simply freeze all the variables, say, in the middle of the iterations, and pick eta as the top 10% of the variable values. This is not a standard end-to-end optimization either.
> > > > > >
> > > > > > End-to-end optimization means one has a loss and just uses backpropagation to update all the parameters; however, here a non-BP operation, like eigenvalue decomposition or picking a threshold in the variables, are injected. It may affect the convergence.
> > > > > >
> > > > > > I have updated my rating to 5 since authors have clarified the notations and evaluation metrics. I feel that the writing should be more rigorous for readers to clearly understand the algorithm and its procedure.

---

> > > > > > > ### Author Response · Authors · 2020-11-25
> > > > > > > **Thank you for your discussion and some further comments**
> > > > > > >
> > > > > > > Dear Reviewer2,
> > > > > > >
> > > > > > > We really appreciate your comments and thank you very much for updating the score. Your input has helped us to make this paper into a better shape, which is indispensable for this paper to meet the high conference standard.
> > > > > > >
> > > > > > > We apologize for not referring “end-to-end” in a fully rigorous way, and we would like to comment a little bit more about our model not being strictly end-to-end.
> > > > > > >
> > > > > > > Thank you for bringing up that spectral clustering is associated with the normalized min cut loss. We totally agree that this is a great point and potential future work. We actually thought about using this loss term, but we didn’t choose to do so since our model requires a hard assignment of each node. We need to decide which node should be included in the sampled subgraph, because we are doing sampling rather than just clustering. Soft-assignments are not sufficient for successive motif learning nor contrastive learning. If we use a continuous relaxation of spectral clustering, say the normalized min cut loss, we will need another discretization step to convert soft assignments to hard assignments. This could include extra parameters like a cutting threshold, which in our opinion is less principled, and the whole framework is still not strictly end-to-end.
> > > > > > >
> > > > > > > For the choice of $\eta$. It could have been set as a hyperparameter, e.g. 0.5, as we only use it to cut cosine similarities. However, we feel that dynamically computing $\eta$ is a more principled way for selecting a proper threshold rather than tuning a hyperparameter. The advantage of our design is that as the motif-to-subgraph similarities change, $\eta$ can adjust its value accordingly, so we won’t update too few or too many embeddings each time (none or all embeddings in the extreme case). Besides, the purpose of $\eta$ is more about controlling which embeddings we want to update, but not about how to do the update. We feel that a non-BP $\eta$ is not harmful in this case.
> > > > > > >
> > > > > > > This not-strictly end-to-end style appeared in other works as well. Like in VQ-VAE [1], in each round, the model decides which embedding it needs to update by a non-differentiable arg-min operation (equation (2)). Also, a stop-gradient operation is used to construct labels for guiding the embeddings to update (equation (3)). None of these non-BP operations prevent VQ-VAE from being a milestone work for learning discrete representations. In our case, dynamically choosing $\eta$ as the top 10% is like a generalization of the arg-min operation in VQ-VAE. The use of spectral clustering for providing labels also have the same flavor as using the stop-gradient operation to provide labels. Compared to the image learning in VQ-VAE, we think it is even harder for discrete data structures like graphs to avoid discretization.
> > > > > > >
> > > > > > > In summary, we agree that our method is not strictly end-to-end. There are possible manipulations we could try, and some of them may potentially improve the model performance. However, our main contribution is the idea of learning motifs, and leveraging these motifs for contrastive learning. For the motif learning part, it turns a combinatorial counting problem into a differentiable learning problem, which makes it possible to generalize to large datasets. For the contrastive part, to the best of our knowledge, no previous works have tried subgraph-to-graph contrastive learning, because figuring out how to pick the right subgraph is non-trivial. Naively using structured-based-only sampling methods has incremental performance gain as shown in our ablation study.
> > > > > > >
> > > > > > > We thank the reviewer again for discussing with us back-and-forth. These meaningful discussions help a lot for us to improve our paper and shine a light on some future work.
> > > > > > >
> > > > > > > [1] van den Oord, A., Vinyals, O., et al. Neural discrete representation learning. In Advances in Neural Information Processing Systems, pp. 6309–6318, 2017. https://arxiv.org/pdf/1711.00937.pdf

---

> ### Author Response · Authors · 2020-11-14
> **Question Answering**
>
> Dear Reviewer2,
>
> We really appreciate your questions and comments. We feel there might be some misunderstanding of our paper and hope our response can address your concerns and clarify our method and contribution.
>
> We are sorry for the confusion caused by the notations. The paper is updated with more precise notations, especially in section 3.  Modifications are highlighted in red. For clarity, pseudocode of the main algorithm and a table summarizing all notations have been included in Appendix A and B.
>
> Regarding the questions you raised. Please see our answers below.
>
> Q: The grouping part and the contrastive part may conflict because subgraphs are shared among different graphs
>
> A: We want to emphasize that our encoding procedure is to first apply GNN over the whole graph to get node embeddings, and then do subgraph pooling, so the subgraph embeddings are contextualized, i.e. determined not only by its own structure, but also its co-occurring neighbor subgraphs.
>
> We use such contextualized embedding design because the semantic property of each subgraph is not only determined by its own structure, but also by its neighborhood (context). For example, in the text domain, the same word “bank” has different meanings when it appears in “bank account” v.s. in “river bank”. Similarly in the graph domain, e.g. for molecules, properties of a functional group are also influenced by other atoms around it. Therefore, our contrastive learning and grouping are designed specifically to make the GNN capture high-order contextual information, and does not lead to confliction. Below we explain why this is the case for both contrastive learning and grouping.
>
> For contrastive learning, we agree that choosing negative samples is non-trivial, and false-negative pairs will cause contrastive learning to fail. Therefore, we adopt this global-message-passing-and-local-aggregating strategy to generate contextual subgraph embedding.
>
> Let me illustrate how this works through an example. Consider two whole graphs $G_1$ and $G_2$. We denote a sampled subgraph from $G_1$ as $g_1$, and remaining nodes not included in $g_1$ as  $\neg g_1$, i.e. $g_1 \in G_1$, $\neg g_1 \in G_1$, and $g_1 \cup \neg g_1 = G_1$. Similarly, we denote a sampled subgraph from $G_2$ as $g_2$, and $\neg g_2 \cup g_2 = G_2$. Let’s further assume two subgraphs g1 and g2 have similar structures, which reflects the “conflict of shared subgraph” in your question.
>
> To generate the subgraph embeddings $e_1$ for $g_1$, we first encode $G_1$ via GNN message passing to get contextualized embedding for each node in $G_1$, and then aggregate embeddings of nodes belong to $g_1$ to get $e_1$. This embedding strategy allows $e_1$ to encode both the structural information of $g_1$ as well as its contextual information from $\neg g_1$. The subgraph embedding $e_2$ for $g_2$ is generated similarly. Therefore, even if $g_1$ and $g_2$ share similar structures, their embeddings $e_1$ and $e_2$ are different due to the different contexts $\neg g_1$ and $\neg g_2$. Afterwards, when using $e_1$ as a positive sample and $e_2$ as a negative sample of $G_1$ to conduct contrastive learning, the model will need to learn higher-level contextual information to distinguish these two subgraphs. $\textbf{This is not a conflict but what we want the model to learn}$.
>
> Negative samples under this setting are helpful for contrastive learning. A comparison would be negative samples with an extra filter. More specifically, before using $g_2$ as a negative sample of $G_1$, we first check whether it shares similar motif assignments as $g_1$. If that is the case, we don’t use $g_2$ as a negative sample to avoid the “conflict” you mentioned. In our experiment, such a procedure actually caused the model performance to drop, which shows that we should encourage the GNN to learn contextualized structural information.
>
> Our design choice of contextualized subgraph embeddings is also crucial for the grouping (motif clustering) part. It allows us to identify semantically similar subgraphs based on similar contexts, even subgraphs themselves can have different structures. For example, for molecules,  a sulfate ($-SO_4$) and a phosphate ($-PO_4$) are two different functional groups with very similar chemical properties. Traditional discrete matching algorithms will treat them as different subgraphs, while our contextualized embeddings and motif learning can help encode them to two close embeddings, as they share very similar contexts, indicating they play similar roles in different molecules.
>
> To summarize, our model design and pre-training objective together will encourage GNNs to encode subgraphs based on their contexts, so it can capture high-order semantic information. Under such an assumption, two structurally-similar subgraphs can have different embeddings, while two structurally-different but contextually-similar subgraphs can have similar embeddings.

---

### Official Review · AnonReviewer3 · 2020-10-28
**Review comments to Paper 2787**

**Rating:** 5
**Confidence:** 4

**Review:**

==========Summary==========

In this work, the authors study how to leverage motif discovery to learn graph representations. MICRO-Graph is proposed to enable simultaneous motif discovery and graph representation learning. Empirical results on public benchmark datasets suggest the effectiveness of the proposed method.

==========Reason for the rating==========

At this moment, I am leaning to reject. Overall, the technical quality is my main concern. This paper's presentation also makes its technical details confusing. Hopefully, the authors could address my concern in the rebuttal.

==========Strong points==========

1. The authors investigate the problem of graph learning from a unique angle of motif discovery.

2. The authors propose the MICRO-Graph framework that enables simultaneous motif discovery and graph representation learning.

3. From multiple public benchmark datasets, the empirical results suggest the proposed technique could be promising in graph classification tasks.

==========Weak points==========

1. My major concern is on the technical quality of this paper.
    - Graph representations generated by a node embedding system may not be comparable across different graphs. A node embedding system is assumed to be available in MICRO-Graph. In most cases, a node embedding system is trained in an unsupervised manner. The generated node representations usually only work in a transductive setting such that the learned model may not be able to be generalized for unseen graphs. In other words, node embedding from different graphs may not be comparable in a meaningful way, although they are vectors of numerical values. For the input node representations, the authors may need to clarify how the node representations are obtained, and why they are comparable across different graphs.
    - The authors may need to clarify whether MICRO-Graph could tune the parameters in the assumed node embedding system. If the parameters in the embedding system are trainable, the authors may need to discuss more on the assumed embedding system and how the parameters impact node representations. If the parameters in the embedding system are fixed, the subgraph segmenter basically generates fixed subgraphs by spectral clustering, and it is difficult to see the point of Equation (6). In sum, without clarification, the technical discussion is confusing.
    - For the motif discovery discussed in 3.2, it seems to be a typical clustering problem. The authors may need to clarify what the unique aspect is in the proposed method.
    - For the discussion in 3.4, it is hard to see why it is reasonable to use subgraph relations to define positive/negative sets, as the difference between a graph and its subgraph could be significant in many cases.
    - In section 4.1.1, the term "fine-tune" is mentioned. Could the authors provide more details on fine-tuning?

2. It is still unclear how motif discovery impacts graph representations. As graph representations are the mean over node representations, the coupling between motif discovery and graph representations seems to be weak.

3. The empirical evaluation could be stronger.
    - For the evaluation in Table 1, "direct supervised learning" should consider existing GNN techniques, such as GraphSage, GAT, GIN, and so on.
    - For the assumed node embedding system, the authors may evaluate how different node embedding systems impact the proposed method.

In addition, the symbol usage in the presentation makes the paper hard to read.

==========Questions during rebuttal period==========

Please address and clarify the weak points above.

==========Post rebuttal==========

I appreciate the authors' great effort on answering my questions. The response clears many confusing points from the original draft. Meanwhile, I still have concerns on how the idea of contrastive learning is handled in this paper, which could have been better shaped. In sum, I have increased the rating accordingly.

---

> ### Author Response · Authors · 2020-11-12
> **Question Answering**
>
> Regarding the questions you raised. Please see our answers below.
>
> Q: Compare representations across different graphs
>
> A: Since we are doing GNN pre-training, we are not explicitly learning a node embedding table for each node, but GNN as a mapping function f is learned. Suppose we want to get node embeddings E given a graph G,  we can use f to generate the embedding, e.g. E = f(G). As long as the node features are in the same domain across different graphs, e.g. molecules share atoms, and papers in citation graphs share words, f will map them into the same space. We don’t consider cross domain training and evaluation, say, pre-train with the citation graph and evaluate on molecules.
>
> Q: Tune node embedding parameters
>
> A: As we described above, we are not learning a node embedding system explicitly. To get the node embeddings, we just embed the node(graph) using the GNN, e.g. E = f(G). Since the parameters of the GNN will be updated,  i.e. f will change, the node embeddings will change as well, and thus "tunable" in this sense. Our loss function, including equation (6), is meant to provide learning signals for the GNN parameters to update.
>
> Q: The unique aspect of the proposed method
>
> A: Simple clustering only finds clustering labels of fixed data points. In the case of representation learning, each point we want to cluster is a representation and will change during learning. The purpose of clustering here is to guide the representation learning. Clustering guidance has been shown to be helpful for representation learning[7,8]. At the same time, we want to keep the representations from degenerate solutions, e.g. collapse to a single point, which is a harder task than simple clustering.
>
> Q: Subgraph relation for positive/negative sets
>
> A: For a whole graph G1, we first use the GNN to do message passing on the G1 and generate all the contextualized node embeddings. Both the embedding of G1 and a sampled subgraph S1 are aggregations of those node embeddings. During the whole graph message passing stage, the nodes within S1 have collected information from other nodes not in S1, so that they contain information of the whole graph G1, and can be treated as a positive sample to G1. On the other hand, a subgraph S2 of a different whole graph G2 has never gone through message passing with any node in G1. Therefore, S2 and G1 form a negative pair.
>
> Q: Details on fine-tuning.
>
> A: Fine-tuning is a conventional evaluation method for a pre-trained model. During the pre-training stage, we train a GNN by a self-supervised task on a large dataset, usually without labels, this GNN can be leveraged to benefit downstream tasks by "fine-tuning". During the fine-tune stage, usually on a smaller supervised dataset, we train the pre-trained GNN using the supervised labels, i.e. we fine tune parameters in it. The high-level idea is that general knowledge of the data has been learned during the pre-train stage. Only a few extra labels are needed to further tune it to adapt to a supervised task. For example on molecule data, during pre-training the GNN has already learned generic knowledge of molecules not restricted to any specific property. During fine-tuning, for questions regarding a specific chemical property, we only need a small labeled dataset to teach the pre-trained GNN about it.
>
> Q: How motif discovery impacts graph representation?
>
> A: As we described above, the motif discovery part will guide the GNN parameters to update, and thus update the node embeddings and further graph embeddings. In particular, equation (6) was meant to make nodes belong to a subgraph to have similar embeddings if this subgraph is similar to a learned motif. We also showed in our ablation study that heuristic sample methods like random walk are hard to generalize, which demonstrates the importance of the motif guidance.
>
> Q: Empirical evaluation.
>
> A: The MICRO-Graph framework is built on GNNs, but not explicit node embedding systems like node2vec. Also, our framework can work with any GNN model architectures, like GraphSage, GIN, and etc. The results shown in the current version are based on the DeeperGCN model.
>
> In general, we summarize our method and highlight our contribution as the following.
>
> 1. Our representation learning is under the pre-train framework, and our contribution is a novel pre-training strategy.
> 2. Besides learning representations, our method can also extract meaning motifs
>
> We hope our answer has made things clear. We are happy for any further discussions or questions, so please feel free to let us know if any of the concerns are not fully addressed.
>
> [7] Caron, M., Bojanowski, P., Joulin, A., Douze, M.: Deep clustering for unsupervised learning of visual features. In: Proceedings of the European Conference on Computer Vision (ECCV) (2018)
>
> [8] Asano YM., Rupprecht C., and Vedaldi A.  Self-labelling via simultaneous clustering and representation learning. In International Conference on Learning Representations (ICLR), 2020

---

> > ### Comment · AnonReviewer3 · 2020-11-24
> > **Response to The authors' response**
> >
> > Hi authors,
> >
> > First of all, I appreciate your hard work on handling our questions. From your response, many confusions have been cleared.
> >
> > On the discussion of "comparable node representations across graphs". The term "node/graph embedding" could be confusing in this context. Alternative terms, such as "node/graph representations", would reduce the ambiguities in the presentation.
> >
> > On the topic of "contrastive learning". I still have doubts on the way to define positive and negative sets. Here, I briefly sketch my understanding.
> > - As the adopted GNNs are inductive, node representations from different graphs could be similar in a meaningful way.
> > - Assume learned node representations within a graph are not over-smoothed. Representations of different nodes within a graph should be discriminative to each other to some extent.
> > - Subgraphs within a graph. Sometimes they should be quite different, especially when there are multiple motifs existing in this graph.
> > - Subgraphs from different graphs. Whether subgraphs are similar or different could mainly depend on how many motifs the original graphs share.
> >
> > To this end, I still feel the proposed way to define positive and negative sets based on which original graph a subgraph is from could be counter intuitive. The following cases could be common.
> > - Given two graphs, they could share many motifs. Therefore, subgraphs from the two graphs could be quite similar.
> > - Given one graph with multiple motifs, their subgraphs could be dissimilar enough so that the discovered motifs are meaningful.
> >
> > Let me know if I miss anything or have any misunderstanding. I'm happy to have further discussions with the authors on the issue around "contrastive learning".

---

> > > ### Author Response · Authors · 2020-11-25
> > > **The false-negative problem and multi-view graph representations**
> > >
> > > Dear Reviewer3,
> > >
> > > We appreciate your questions and comments. We hope our response can address your concerns.
> > >
> > > Thank you for pointing out the ambiguity caused by embeddings v.s. representations. We will be careful about our word choice in an updated paper and reduce confusion as much as we can.
> > >
> > > For the first case you mentioned, i.e. two subgraphs sharing many motifs, yes, it is a hard case. This is essentially the false negative problem the whole contrastive learning idea is facing, i.e. different but very similar data instances are utilized to construct negative pairs. The false-negative problem is in not only the graph domain, but other domains like computer vision and language as well, whenever we need to do negative sampling. However, contrastive learning has been shown to be very effective, like word2vec for language and SimCLR for images. That is because most of the time false negatives are rare, and their influence is not statistically significant.
> > >
> > > The false-negative problem is indeed a challenge as you pointed out. In our experiments, we find that false negatives heavily influence $\textbf{subgraph-to-subgraph}$ contrastive learning, because it is very likely for two subgraphs from different whole graphs to be similar. Therefore, we choose to design our contrastive learning as $\textbf{graph-to-subgraph}$. It alleviates the false negative problem because it is less likely for two whole graphs to share many common subgraphs. We are not claiming false negatives are uncommon in this case, but they are not statistically significant compared to many true negatives we have, and the model can still learn from contrastive prediction.
> > >
> > > We also provide empirical results for our claim above. The claim is saying that the graph dataset is well distributed and contains diverse whole graphs. In our newly added visualizations in Appendix H, we show some statistics of the dataset to assess how likely two graphs will have many common motifs. Specifically, we first construct the graph-motif-assignment vector $a_i$ for each graph $G_i$ with $N_i$ subgraphs $g_i, …, g_{N_i}$. To get $a_i$, we aggregate the normalized motif-subgraph similarities $\tilde S$ as the following.
> > >
> > > $a_i = \frac{1}{N_i}\sum_{g_j \in G_i} \tilde S_{:, j}$
> > >
> > > Each $a_i$ is a K-dimensional vector representing which motifs the graph $G_i$ contains. We then take all the $a_i$’s and compute their pairwise cosine similarities. In the dataset we use, only 8.7% of these pairwise cosine similarities are higher than 0.5, and only 1.7% are higher than 0.9. Moreover, for a skewed dataset containing mostly only similar graphs, we can add an extra filter based on these graph-motif-assignment vectors $a_i$’s. In other words, if two graphs $G_1$ and $G_2$ have very similar $a_1$ and $a_2$, we won’t use these two graphs to construct negative pairs. However, since this filter is not the main contribution of our work, and even without this filter our model achieves good results, we will leave investigation of proper filters as a future work.
> > >
> > > For the second case you mentioned, i.e. motifs from the same whole graph should be allowed to have different representations, yes, it is also what we desired, and it is what our model is capable of learning. In our newly added Appendix H, we show more visualizations of the learned motif representations and whole graph representations. Basically, we show that $\textbf{for the representations our model learned, two motifs can be similar to the same whole graph, while different from each other}$.
> > >
> > > The intuition is that a graph representation is a multi-view representation. As we are using relatively high-dim vectors (300-dim), the graph representation can contain information from multiple motifs in different dimensions. Two different motifs can both have high similarity scores to a whole graph, because different dimensions of the whole graph representation are activated. In other words, two different motifs are actually similar to two projections of the whole graph on different basis. This is an observation we make from the visualizations we show in Appendix H.
> > >
> > > We appreciate your input in making this paper into a better shape, please feel free to ask any more questions or make any more comments. It’s our pleasure to have further discussions.

---

> ### Author Response · Authors · 2020-11-13
> **Paper approach clarification**
>
> Dear Reviewer3,
>
> We appreciate your questions and comments. We feel that there might be some misunderstanding of our paper, and we hope our response can address your concerns and further clarify our method and contribution.
>
> We are sorry for the confusion caused by the symbols. The paper is updated with more precise symbols, especially in section 3. Modifications are highlighted in red. For clarity, the pseudocode of the main algorithm and a table of symbols are now in Appendix A and B.
>
> The representation learning method in MICRO-Graph is under the framework of GNN pre-training. Under this framework, a GNN model is trained with a self-supervised task on the pre-train dataset. The pre-trained GNN can be used to inductively generate node and graph embeddings of unseen graphs. The idea of pre-training has been shown to be effective in the graph domain[1,2], as well as other domains like language[3,4] and computer vision[5,6].
>
> Given a graph with node features X and adjacency matrix A, A GNN is meant to be a function f that maps the input graph to node embeddings E, i.e. E = f(X, A). We are learning the function f by updating parameters within the GNN, and we are not learning the embeddings E explicitly.
>
> The GNN-based representation learning framework is inductive by its nature, since GNNs can generate embeddings based on the local neighborhood information.  A pre-trained GNN is a function f that captures the generic knowledge. Even for unseen graphs, as long as node features (atoms types of molecules, like carbon, oxygen, and nitrogen) and local structures (bonds between atoms) are given, GNNs can map nodes from different graphs into the same space. As a comparison, traditional node embedding learning algorithms like node2vec and DeepWalk will maintain an embedding table E containing each node in the training dataset and update the table directly. Therefore, they cannot handle unseen nodes or unseen graphs. GNNs are not restricted by the embedding table. For example, we can pre-train a GNN on a set of molecules. When an unseen molecule comes, because the pre-trained GNN has seen atoms and bonds of this unseen molecule in other molecules before, it can infer the embeddings of nodes in this new molecule and conduct downstream tasks.
>
> Reference
> [1] Weihua  Hu,  Bowen  Liu,  Joseph  Gomes,  Marinka  Zitnik,  Percy  Liang,  Vijay  Pande,  and  Jure Leskovec. Strategies  for  pre-training  graph  neural  networks. In International  Conference on  Learning  Representations,  2020b
>
> [2] Ziniu Hu, Yuxiao Dong, Kuansan Wang, Kai-Wei Chang, and Yizhou Sun.  Gpt-gnn:  Generativepre-training of graph neural networks.  InProceedings of the 26th ACM SIGKDD Conference on Knowledge Discovery and Data Mining, 2020c.
>
> [3] Jacob Devlin, Ming-Wei Chang, Kenton Lee, and Kristina Toutanova. BERT: Pre-training of deep bidirectional transformers for language understanding. In Annual Conference of the North American Chapter of the Association for Computational Linguistics (NAACL), 2019.
>
> [4] Alec Radford, Karthik Narasimhan, Tim Salimans, and Ilya Sutskever. Improving language understanding by generative pre-training. 2018.
>
> [5] Ting Chen, Simon Kornblith, Mohammad Norouzi, and Geoffrey Hinton.  A simple framework for contrastive learning of visual representations, 2020
>
> [6] Kaiming  He,  Haoqi  Fan,  Yuxin  Wu,  Saining  Xie,  and  Ross  Girshick.   Momentum  contrast  for unsupervised visual representation learning. arXiv preprint arXiv:1911.05722, 2019

---

### Official Review · AnonReviewer4 · 2020-10-28
**Review of MICRO**

**Rating:** 5
**Confidence:** 5

**Review:**

Overall:
This paper proposes an interesting framework
+ It extracts subgraph(s) for each graph from node affinity matrix and spectral clustering, together with the help of motifs.
+ It learns the motifs by clustering the subgraphs.
+ It applies contrastive self-supervised learning on the graph-subgraph pair.
+ It overcomes the combinatorial problem by learning the motifs on the continuous representation space.

Strengths:
+ The idea is novel and seems promising.
+ This paper is technically correct.
+ Nice visualization of the learned motifs.

Weaknesses:
+ This paper is not well written, especially the notations. I list the main concerns here, and hope the authors can help clarify them later.
1. S3, N is not defined, and M is defined as #graph. But some following sections are implying N as #graph and M as #subgraph.
2. S3.2, should be '... we can extract M subgraphs …'
3. S3.2, according to 'sampling a subgraph from a graph', does this mean each graph has one and only one subgraph?
4. S3.2, ‘... apply the subgraph index as a mask to its subgraph embedding …’ I can understand the following equation but not this sentence.
5. S3.2, should be '{..., m_K}'
6. Actually without clarifying the above detailed notations, it’s a little hard to follow the remaining (sub)sections. e.g., what are S and Q? Because M is #subgraph, and it should be [..., S_M] and [..., q_M] according to the descriptions.
7. S3.3, {g_{i,j}}_{j=1}^M, I guess this is saying each graph can have multiple groups/subgraphs and such mapping is represented in the N*M binary matrix. Then it contradicts with the (3) mentioned above, with 1-1 mapping between the two views.
8. In Eq 5, not sure if softmax_s matches with the description above.
9. In Eq 6, it would be better to add ‘1 \le k \le K’ in the last term.
+ Experiments, authors can consider adding more baselines.
1. GROVER [1] is the SOTA, where it randomly masks a subgraph, which is highly relevant to this paper.
2. GNNExplainer [2] is learning the motif in an end-to-end way, the authors could also consider comparing with it.

Recommendation:
Considering the notation issues listed above and lack of baselines, I would encourage the authors to polish up this paper. It has the potential to be a much better paper. For now, I would reject the current version.

Questions:
+ OGB[3] was first released in May 2020, and GROVER[1] was released one month after that. So I think at least the authors should cite [1].
+ In S1, ‘Previous approaches, such as …’ Deep Graph Informax and InforGraph, in the last layer it is indeed node-to-graph views, but if we take it under the GNN setting, where each node representation in the last layer actually encodes a K-hop neighborhood around that node, then it is subgraph-to-graph views.
+ The ContextPred in Figure 2 is not correct: it should be contrasting the k-hop neighborhood and pre-defined context graph. Check Figure 2 in [4].
+ In Table 1, the ContextPred is the worst, which is not expected based on my own experience, any reason why? I couldn’t find it in the supplementary files.
+ Comparing Table 1 and 2, the frozen pre-trained GNN seems to be much better to the fine-tuning ones. Can authors discuss this further?
+ Figure 3 to 6 are blurry.


[1] https://arxiv.org/abs/2007.02835

[2] https://arxiv.org/abs/1903.03894

[3] https://arxiv.org/abs/2005.00687

[4] https://arxiv.org/abs/1905.12265

---

> ### Author Response · Authors · 2020-11-17
> **Question Answering Continued**
>
> Q: Counter-intuitive experimental results in Table 1 and Table 2
>
> A: We are sorry for the confusion caused by the fine-tune evaluation and linear evaluation. The original Table1 and Table2 in the paper were not directly comparable, as they were meant to compare with two previous works [1] and [2]. Therefore, we closely followed the settings of these two works respectively, where the evaluation metric of Table1 was AUC-ROC and the evaluation of Table 2 was accuracy. We have now updated Table2, so both cases use AUC-ROC. The old version of Table2 has been moved to Appendix. A copy of Table2 is shown below.
>
> |SSL methods   | bace              | bbbp             |clintox           |hiv                 |sider              | tox21            |toxcast           |Avg|
> |-------------------|--------------|--------------|----------------|----------------|----------------|----------------|----------------|----------------|
> |ContextPred    | 53.09 ± 0.84 | 55.51 ± 0.08 | 40.73 ± 0.02 | 53.31 ± 0.15 | 52.28 ± 0.08 | 35.31 ± 0.25 | 47.06 ± 0.06 | 48.18 |
> |InfoGraph        | 66.06 ± 0.82 | 75.34 ± 0.51 | 75.71 ± 0.53 | 61.45 ± 0.74 | 54.7 ± 0.24   | 63.95 ± 0.24 | 52.69 ± 0.07 | 64.27 |
> |GPT-GNN         | 59.43 ± 0.66 | 71.58 ± 0.54 | 62.78 ± 0.58 | 64.08 ± 0.36 | 54.67 ± 0.16 | 68.2 ± 0.14   | 57.06 ± 0.13 | 62.53 |
> |GROVER|65.67±0.38|78.47±0.36|53.19±0.68|69.03±0.23|54.94±0.12|67.63±0.13|57.28±0.05| 63.74|
> |MICRO-Graph | 69.54 ± 0.39 | 81.07 ± 0.42 | 63.69 ± 0.56 | 72.74 ± 0.15 | 55.39 ± 0.26 | 72.91 ± 0.12 | 61.04 ± 0.07 | 68.05|
>
> The comparison between the updated linear evaluation and fine-tune evaluation is consistent with our expectation (and also yours~). As you pointed out, linear evaluation results should be lower than the fine-tune results. In particular, MICRO-Graph’s linear evaluation result is 5.14% lower than fine-tune on average, and for baselines, the average linear results are lower by 22.47%, 7.84%, 9.65%, and 8.76% respectively. This shows MICRO-Graph has learned more robust representations during pre-training, and a simple linear classifier on top can achieve good performance.
>
>
> Q: Views of Deep Graph Infomax and InfoGraph in Figure 2.
>
> A: Yes, we totally agree that for both DGI and InfoGraph, their last layer node representations contain subgraph-level information. However, in Figure 2, what we want to emphasize is the view construction difference rather than the representation difference. How we construct views is an important difference between MICRO-Graph and others. Specifically, we construct views by pooling nodes from a motif-guided subgraph sample, while others rely on single nodes. Like in computer vision, when input an image into a CNN, each pixel representation in the last layer corresponds to not a single pixel but a receptive field. However, the SOTA approach is using a group of pixels to represent a region in the image rather than using a single pixel. The difficulty of adopting similar approaches on graphs is to figure out which nodes we should group, i.e. which nodes form a meaningful subgraph. This is exactly our contribution, i.e. the motifs discovered by our framework. Models like InfoGraph don’t utilize this information.
>
> Q: Incorrect illustration of ContextPred in Figure 2
>
> A: Thank you for pointing this out, we have updated Figure 2. We would like to emphasize that even though the figure was incorrect, our implementation is taken from the official code by the paper authors. Therefore, our experiment result is still valid. In the next question we discuss more about it.
>
> Q: ContextPred result is lower than other models
>
> A: ContextPred has lower performance because it is compared to strong baselines, which are concurrent or later work to ContextPred and not compared in the original ContextPred paper. Our ContextPred implementation and hyperparameters are taken from the official code and paper. To eliminate possible influence due to different hyperparameters, we add two things in the Appendix. 1) a detailed description of hyperparameters used for ContextPred 2) Additional experiments with ContextPred using different hyperparameters. However, we didn’t observe significant differences between these results and the results in Table 1 and Table 2.
>
>
> Q: Figure 3 to 6 are blurry.
>
> A: These figures are updated.
>
> We hope our answer has made things clear. We are happy for any further discussions or questions, so please feel free to let us know if any of the concerns are not fully addressed.
>
>
> Reference
>
> [1] Weihua  Hu,  Bowen  Liu,  Joseph  Gomes,  Marinka  Zitnik,  Percy  Liang,  Vijay  Pande,  and  Jure Leskovec. Strategies  for  pre-training  graph  neural  networks. In International  Conference on  Learning  Representations,  2020b
>
> [2]Fan-Yun Sun, Jordan Hoffmann, Vikas Verma, and Jian Tang. Infograph: Unsupervised and semi- supervised graph-level representation learning via mutual information maximization, 2019.

---

> > ### Comment · AnonReviewer4 · 2020-11-18
> > **Reply to Authors**
> >
> > Hi Authros,
> >
> > I saw you answer most of my questions and did a lot of work during the rebuttal, and I do acknowledge this. BTW. I notice you also update Figure 2, Figure 3, and Figure 6, which should also be marked?
> >
> > Below are some of my thoughts:
> > (1) I think this paper is a nice empirical work. This is a well-motivated work and applies the important substructure (motif) idea in graph neural networks.
> > (2) I saw the authors updated most parts of the `Method Section`  and some parts of the `Experiment Section`.  For the previous `Method Section`, I would say it is indeed causing huge problems instead of some small typos. Similarly, for the `Experiment Section`,  the newly-added ablation studies, like GROVER, are important but missing in the first version.
> > (3) Plus the point (2) and my initial reviewing comments (like the mis-understanding of important baselines), I don't think the original submission made a clear and robust story to the audience. Considering that this is an empirical work, which should present a nice and solid story through careful experiments, plus clean and clear method descriptions, I made the previous score. And I think other reviewers also had the same opinions on this.
> > (4) As mentioned in (2), the key components of this empirical work were either incorrect or missing. Now the new version fixes them and looks much better. But to me, it is more like a `rewriting` instead of `revision`.
> >
> > So combining the previous comments, and to be fair to other authors, now I would like to hold the original score.
> >
> > I'm very happy for further discussions on this.
> >
> > Again, I appreciate the authors' hard work in rebuttal.

---

> > > ### Author Response · Authors · 2020-11-19
> > > **Reply to Reviewer4**
> > >
> > > Dear Reviewer4,
> > >
> > > We appreciate your further comments. Your meaningful and constructive reviews really help us to improve the quality of our paper.
> > >
> > > We have indicated “the figure is updated” in the figure caption 2 to 6. We apologize again for the confusion caused by the original version. We believe we have made corresponding updates in the paper and answered all of your questions in our previous response. If there are other specific concerns, we are happy for further discussions.
> > >
> > > However, we wouldn’t agree that our updated version of the paper is rewriting, given that the methodology remains the same. For the method section in particular, although we highlighted it in red, we actually only polished our writing for clarity. Maybe the red highlight was too much for reading. We just changed it to very accurately reflect the important changes. We would like to emphasize that our storyline and algorithm both stay unchanged. More specifically, we have made the following updates.
> > >
> > > 1. Clarify N as the number of subgraphs and M as the number of graphs
> > >
> > > 2. Clarify $\boldsymbol h_i$ as the embedding of whole graphs and $\boldsymbol e_j$ as the embedding of subgraphs
> > >
> > > 3. Clarify more than one subgraphs are generated from each whole graph
> > >
> > > 4. For notation clarity, introduce a new notation $\boldsymbol{\tilde S}$ as $\boldsymbol S$ after softmax normalization
> > >
> > > 5. Use superscripts instead of multiple subscripts for the node-node affinity matrix $\boldsymbol A^{(i)}$
> > >
> > > 6. For notation clarity, introduce a new notation $\mathbb I$ as the index set for nodes selected as a subgraph
> > >
> > > 7. Some word changes and new concrete examples for better illustration
> > >
> > > 8. An algorithm in pseudocode, along with a paragraph going over each line of the code
> > >
> > > None of these changes affect our methodology or conclusion. We didn’t make these changes to claim more contribution either. We did it only for better convey the idea.
> > >
> > > The fix of our figures is also for illustration purposes. Besides, there are clear explanations of the baseline models when we introduce them in the experiment section.
> > >
> > > Regarding the missing baseline. We totally agree that GROVER is an interesting work and an important baseline with a strong performance. We are sorry that we weren’t aware of this work before our original submission, but we think adding new baselines is common for the reviewing process of any conferences. Moreover, we have included a detailed discussion of GROVER’s contribution in the related work section.
> > >
> > > The merit of ICLR is its publicly open reviewing process, which distinguishes ICLR from other conferences. It encourages the improvement of papers involving reviewers in the loop. In our case, we didn’t change the methodology, but only revised our paper according to suggestions given by all the reviewers. We followed the policy of ICLR, and we appreciate a fair judgment regarding our updated version, rather than punishment regarding the initial version.
> > >
> > > We also agree that it is important to be fair for all submissions. However, all the paper authors get an equal chance to update their own work during the rebuttal period. To really be fair to everyone, we are leaning to a judgment based on the revised paper for all authors, as long as the methodology stays unchanged. We welcome any discussion if anything is unclear in the current version. We also sincerely appreciate your inputs in making this paper into a better shape, which is indispensable for this paper to meet the high conference standard.
> > >
> > > Again, please feel free to ask any more questions or make any more comments. It’s our pleasure to have further discussions.

---

> ### Author Response · Authors · 2020-11-17
> **Question Answering**
>
> Dear Reviewer4,
>
> We appreciate your questions and comments. We hope our response can address your concerns.
>
> We are sorry for the confusion caused by the notations. The paper is updated with more precise notations, especially in section 3.  Modifications are highlighted in red. For clarity, pseudocode of the main algorithm and a table of all notations have been included in Appendix A and B.
>
> Q: GROVER baseline
>
> A: Thank you for introducing the interesting GROVER model. We have added GROVER in our experiment. The GROVER paper proposes a Transformer model and a pre-train task. Since our focus is the pre-train task that generalizes to all models, our experiment applies the pre-train task in GROVER to the DeeperGCN. This makes the comparison fair since MICRO-Graph and other baselines all use DeeperGCN. Table1 and Table2 are updated to include GROVER.
> We would like to emphasize that the pre-train task in GROVER needs discrete-counting-based motif extraction by professional software. This step requires domain knowledge and cannot generalize to graphs with high-dimensional or continuous features. This is exactly what our paper tries to overcome.
>
> Fine-tune (Table1)
>
> |SSL methods   | bace              | bbbp             |clintox           |hiv                 |sider              | tox21            |toxcast           |Avg|
> |-------------------|--------------|--------------|----------------|----------------|----------------|----------------|----------------|----------------|
> Non-Pretrain | 72.80 | 82.13 | 74.98 | 73.38 | 55.65 | 76.10 | 63.34  | 71.19
> ContextPred | 73.02 | 80.94 | 74.57 | 73.85 | 54.15 | 74.85 | 63.19  | 70.65
> InfoGraph | 76.09 | 80.38 | 78.36 | 72.59 | 56.88 | 76.12 | 64.40 | 72.11
> GPT-GNN | 75.56 | 83.35 | 74.84 | 74.82 | 55.59 | 76.34 | 64.76 | 72.18
> GROVER | 75.22 | 83.16 | 76.8 | 74.46 | 56.63 | 76.77 | 64.43|72.5
> MCRIO-Graph | 76.16 | 83.78 | 77.50 | 75.51 | 57.28 | 76.68 | 65.42 | 73.19
>
> The result shows that GROVER is a strong model. It outperforms three baselines, but not as good as MICRO-Graph. Although both are motif-based self-supervised learning, it is reasonable that MICRO-Graph is better. This is because motifs in GROVER are discrete structures, but motifs in MICRO-Graph are contextualized embeddings. Let me illustrate why this difference matters through an example.
>
> Consider two graphs $G_1$ and $G_2$ that share the same motif in the GROVER sense. We denote this motif in $G_1$ as $g_1$, and remaining nodes not in $g_1$ as  $\neg g_1$, i.e. $g_1 \in G_1$, $\neg g_1 \in G_1$, and $g_1 \cup \neg g_1 = G_1$. Similarly, we denote this motif in $G_2$ as $g_2$, and $\neg g_2 \cup g_2 = G_2$.
>
> For GROVER, the model learns by predicting $g_1$ belongs to $G_1$ and $g_2$ belongs to $G_2$. Only information within $g_1$ and $g_2$ are used for constructing labels. Information from other nodes are ignored. In this case, GROVER cannot distinguish $G_1$ and $G_2$ by pre-training on these labels.
>
> For MICRO-Graph, we generate the subgraph embedding $e_1$ for $g_1$. We first encode $G_1$ via GNN message passing to get contextualized embedding for each node in $G_1$, and then aggregate embeddings of nodes belong to $g_1$ to get $e_1$. This allows $e_1$ to encode both the structural information of $g_1$ and contextual information from $\neg g_1$. The subgraph embedding $e_2$ for $g_2$ is generated similarly. Therefore, even if $g_1$ and $g_2$ are the same motif, $e_1$ and $e_2$ are different due to the different contexts $\neg g_1$ and $\neg g_2$. When using $e_1$ as a positive sample and $e_2$ as a negative sample of $G_1$ for contrastive learning, the model will need to learn higher-level contextual information to distinguish these two subgraphs. Thus, the model learns more than what is in the subgraph, but also contextual information extracted from other nodes.
>
> Q: GNNExplainer baseline
>
> A: Thank you for mentioning GNNExplainer. We would like to discuss the difference between our paper and the GNNExplainer paper. First of all, the task is different. GNNExplainer focuses on post-process model interpretation. For any fixed model, GNNExplainer explains which part of the input graph is important for the prediction, and the analysis is done at a single-graph level. In contrast, MICRO-Graph trains a model and discovers universal motifs in the whole dataset. Secondly, the setting is different. For GNNExplainer, GNN models to be explained are under the supervised setting. The definition of important substructures is $\textit{high mutual information with the supervised task prediction}$. In MICRO-Graph, the learning process is self-supervised. The way we define important substructures in MICRO-Graph, i.e. motifs, is $\textit{frequently occurring patterns}$. Both papers try to learn important substructures of graphs. However, given these differences, they are not directly comparable.
>
> The discussion of GROVER and GNNExplainer is added to the related work, and these two papers are cited accordingly.

---

### Official Review · AnonReviewer1 · 2020-10-29
**Review for Paper 2787**

**Rating:** 6
**Confidence:** 3

**Review:**

Paper Summary

The paper describes a self-supervised framework to extract graph motifs and use them as input for downstream contrastive learning. The framework contains three components: (a) motif guided segmenter to derive node subgraphs, (b) a motif learning - a clustering task among the subgraphs to identify concrete graph motifs and (c) contrastive learner for downstream graph tasks. The global objective is defined as the sum of the likelihoods of the three components.  The framework is evaluated using from a large scale chemical compound graph dataset. The evaluation is performed for both transfer learning and utility of extracted features and outperforms the tested competing methods.


Positives:
* The framework presented to identity graph motifs and then use of learned motifs in contrastive learning for graph representations is very interesting and does lead to substantial gains in performance at least in the datasets tested
* Experimental design to evaluate the framework is well thought too to specifically test the different pieces and components.
* The results with synthetic dataset was a nice addition and a more thorough treatment with quantification of results, particularly recovery of true motifs would be a good addition.
* The paper is generally well written. An algorithm/pseudo-code in the supplement would have made it even easier to follow given the many moving pieces.


Concerns
* A concern is that the motif learner enforces all clusters to be have similar size. I do not think this is a very realistic assumption in real world datasets.
* The authors should provide some commentary on how the number of clusters used for spectral clustering and what their impact is on downstream results are
* The whole graph embedding on Page 5 uses an average of all node embeddings for the graph. Does this put a limit on the size of the graph itself for the framework to work? If the graph is large, the average node embedding is probably not an accurate representation for the graph ?

---

> ### Author Response · Authors · 2020-11-18
> **Question Answering**
>
> Dear Reviewer1,
>
> We really appreciate your questions and comments. We hope our response can address your concerns.
>
> The paper is updated.  Modifications are highlighted in red. As you suggested, the pseudocode of our algorithm has been included in Appendix A.
>
> Q: Similar-size constraint of clusters
>
> A: We totally agree that strictly forcing all clusters to have the same sizes is an unrealistic assumption. In our case, this constraint is introduced to avoid one cluster being too large that distorts the representation space, which in the extreme case leads to a degenerate solution, i.e. all representations collapse to a single cluster. Our approach can achieve this goal while maintaining relatively flexible cluster sizes.
>
> Theoretically, we use the Sinkhorn-Knopp algorithm to approximate the constraint optimization problem. Equal-partition is not strictly forced, and it is less of a constraint but more of a regularization. It is also shown in [1] Section 3.2 that this regularization can be interpreted as maximizing mutual information between subgraph-index and motif-labels. Therefore, it is reasonable from the information theory perspective.
>
> Empirically, we observe that the final distribution of motif cluster sizes is not uniform. We have included this distribution plot in Appendix G. Also, notice that these clusters do not need to perfectly correspond to the “natural” clusters. As our subgraph embeddings contain contextual information of the whole graph it belongs to, even two subgraphs with the same structure can have different embeddings. In this sense, the clusters are more fine-grained. Large clusters corresponding to the same common subgraph structure can be broken down, resulting in multiple small clusters of the same subgraph structure in different contexts. We thus consider the “overclustering” scheme in our ablation study. As shown in Table 4, we increase the number of clusters from 20 to 100 and observe similar performance for these two cases.
>
> Q: Choosing the number of clusters used in spectral clustering
>
> A: In our case, the number of clusters used in spectral clustering is controlled by a hyper-parameter minNode. minNode describes how many nodes we expect the smallest subgraph sample to have. The cluster number of spectral clustering is set to be $round((graph size / minNode)^{(1/2)})$+ 1. In other words, we choose the number of clusters to be a smooth function of both the graph size and the minNode. In our experiments, we set minNode equal to the smallest graph size in the dataset, which equals 4 for the molecule pre-train dataset. The setting makes sure the number of clusters is always >= 2. A larger minNode value means the number of clusters equals one for small graphs in the dataset, so spectral clustering can’t segment them. On the other hand, we observed that smaller minNode fragments the whole graph and lead to bad results for both motif learning and downstream tasks.
>
> Q: Aggregation for graph embedding
>
> A: How to aggregate node embeddings to get graph embeddings is a very interesting and important question. However, this remains an open problem and it is not the focus of our paper. We would like to point out that our framework can work with any aggregation methods, e.g. mean function, sum function, and entrywise max function. We choose to use the mean function in our experiment, which is the state or the art for empirical performance. We add investigation of aggregation methods to our future work.
>
> We hope our answer has made things clear. We are happy for any further discussions or questions, so please feel free to let us know if any of the concerns are not fully addressed.
>
> Reference
>
> [1] Asano YM., Rupprecht C., and Vedaldi A.  Self-labelling via simultaneous clustering and representation learning. In International Conference on Learning Representations (ICLR), 2020

---

### Public Comment · ~Yuning_You1 · 2020-11-10
**Interesting perspective and related work**

We would like to draw your attention that we have related work demonstrates the power of self-supervision and contrastive learning with augmentations in the graph domain. We hope to have further discussions and references with you.

When Does Self-Supervision Help Graph Convolutional Networks? ICML 2020. https://arxiv.org/abs/2006.09136

Graph Contrastive Learning with Augmentations, NeurIPS 2020. https://arxiv.org/abs/2010.13902

---

> ### Author Response · Authors · 2020-11-17
> **Thank You for Introducing These Interesting Works**
>
> Dear Yuning,
>
> Thank you so much for pointing this out. These are very interesting works and they are related to our paper. We will consider comparing our model with these results in a later version.

---

### Author Response · Authors · 2020-11-25
**Rebuttal revision summary and thanks to all the reviewers and the area chair**

Dear All Reviewers and The Area Chair,

We really appreciate all the questions and comments. These are super useful feedback that helps to make this paper into a better shape. We hope our answers have fully addressed the concerns of all the reviewers.

We have updated our paper. In the latest draft, we have substantially improved our writing to clearly explain all the technical details as reviewers suggested. Important revisions are highlighted in red. We have also included several new parts including

1. New baseline results, e.g. the GROVER model. (Section 4.2)
2. More experiment results on the ContextPred baseline with different hyperparameters (Appendix F)
3. An algorithm in pseudocode (Section 3.5 and Appendix A) and a paragraph going over each line of the code (Section 3.5)
4. A notation summary table (Appendix B)
5. Visualization of motif cluster sizes (Appendix G)
6. Visualization and analysis of various similarity scores, e.g. subgraph-to-subgraph, graph-to-subgraph, and etc. (Appendix H)

We thank you for your time and input again.

All authors

---

### Decision · Program_Chairs · 2021-01-07
**Final Decision**

**Decision:**

Reject

**Comment:**

This paper presents a pre-training strategy for learning graph representations using a graph-to-subgraph contrastive learning objective that also simultaneously discovers motifs. Pre-training for graph representation learning is an important research topic and this work presents a unique solution leveraging the fact that graphs sharing a lot of motifs should be similar to one another. The approach is novel and interesting, the ability to simultaneously identify motifs are highly desirable. The results are promising showing that the proposed approach, when pretrained on the ogbn-molhiv molecule dataset, worked well for several downstream chemical property prediction tasks.

However, the paper is not without weaknesses and the reviewers noticed several of them. There are many parts of the system, the graph segmenter, which relies on spectral clustering (on the affinity matrix), the EM style clustering component to extract the motifs based on the subgraphs,  the sampling loss based on the subgraph-to-motif similarity, and the graph-to-subgraph contrastive learning loss. These parts are tied together through different mechanisms and the training procedure becomes very confusing. It is unclear which parts are updated on the backpropagation path from which loss, and what choices are decided offline (i.e., not integrated into the backpropagation).  This presents great difficulty in understanding and probably using /building-on the method. The paper has improved some aspects of its presentation during the review/discussion process, but the training/optimization procedure of the current version still appears quite opaque, and the reviewers heavily relied on the back and forth discussion to understand what is really going on.

Another concern is that the intuition behind some aspects of the approach and the connections between different components of the approach are a bit difficult to get/digest at places.  The intuition behind graph to subgraph contrastive learning appeared weak to the reviewer. It would be desirable to see a directly comparison to the subgraph-to-subgraph version. The connection between the motif discovery and the representation learning can be somewhat lost as we try to keep the many moving parts straight in the mind.   For these reasons, the paper, in its current form, cannot be accepted.